# A confinable home-and-rescue gene drive for population modification

**Nikolay P Kandul[1], Junru Liu[1], Jared B Bennett[2], John M Marshall[3], Omar S Akbari[1]\***

[1]Section of Cell and Developmental Biology, University of California, San Diego, San Diego, United States; [2]Biophysics Graduate Group, University of California, Berkeley, Berkeley, United States; [3]Division of Epidemiology and Biostatistics, School of Public Health, University of California, Berkeley, Berkeley, United States

**Abstract** Homing-based gene drives, engineered using CRISPR/Cas9, have been proposed to spread desirable genes throughout populations. However, invasion of such drives can be hindered by the accumulation of resistant alleles. To limit this obstacle, we engineer a confinable population modification home-and-rescue (HomeR) drive in *Drosophila* targeting an essential gene. In our experiments, resistant alleles that disrupt the target gene function were recessive lethal and therefore disadvantaged. We demonstrate that HomeR can achieve an increase in frequency in population cage experiments, but that fitness costs due to the Cas9 insertion limit drive efficacy. Finally, we conduct mathematical modeling comparing HomeR to contemporary gene drive architectures for population modification over wide ranges of fitness costs, transmission rates, and release regimens. HomeR could potentially be adapted to other species, as a means for safe, confinable, modification of wild populations.

## Introduction

Effective insect control strategies are necessary for preventing human diseases, such as malaria and dengue virus, and protecting crops from pests. These challenges have fostered the development of innovative population control technologies such as Cas9/guideRNA (Cas9/gRNA) homing-based gene drives (HGDs) (*Champer et al., 2016*; *Esvelt et al., 2014*) which have been laboratory-tested for either population modification (*Adolfi et al., 2020*; *Carballar-Lejarazú et al., 2020*; *Gantz et al., 2015*; *Li et al., 2020*; *Pham et al., 2019*) to spread desirable traits that can impair the mosquitoes' ability to transmit pathogens (e.g. *Buchman et al., 2020*; *Buchman et al., 2019*; *Hoermann et al., 2020*; *Isaacs et al., 2012*; *Marshall et al., 2019*) or population suppression (*Hammond et al., 2016*; *Kyrou et al., 2018*; *Simoni et al., 2020*) to reduce and eliminate wild disease-transmitting populations of mosquitoes. Despite significant progress, HGDs are still an emerging technology that can suffer from the formation of resistant alleles, hindering their efficacy (*Adolfi et al., 2020*; *Carballar-Lejarazú et al., 2020*; *Gantz et al., 2015*; *Hammond et al., 2016*; *Kandul et al., 2020*; *Kyrou et al., 2018*; *Li et al., 2020*; *Pham et al., 2019*; *Simoni et al., 2020*).

In CRISPR/Cas9, the Cas9 endonuclease cuts a programmed DNA sequence complementary to a user-defined short guide RNA molecule (gRNA). To engineer an HGD, leveraging creative designs originally proposed by *Burt, 2003*, CRISPR components are integrated at the target site in the genome. These components are configured so that when they cut the recipient wildtype (*wt*) allele, it is repaired via homology-directed repair (HDR) in heterozygotes, using the donor allele (i.e. allele harboring the HGD) as a template for DNA repair. This enables the HGD to home, or copy, itself into the recipient allele (*Alphey et al., 2020*; *Champer et al., 2016*; *Esvelt et al., 2014*) (referred to as homing from hereon). This general architecture for HGD was quickly adopted, and many HGDs were developed in several insect species (*Gantz et al., 2015*; *Hammond et al., 2016*; *Kandul et al.,*

**\*For correspondence:**
oakbari@ucsd.edu

*2020*; *Kyrou et al., 2018*; *Li et al., 2020*; *Simoni et al., 2020*; *Verkuijl et al., 2020*). However, it soon became widely apparent that HGDs unintentionally promote the formation of resistant alleles through mutagenic repair. When these alleles are positively selected, they can hinder HGD spread in laboratory cage populations (*Champer et al., 2017*; *Hammond et al., 2017*; *Kandul et al., 2020*; *KaramiNejadRanjbar et al., 2018*; *Oberhofer et al., 2018*), with one exception that targeted a conserved sex determination gene for population suppression (*Kyrou et al., 2018*; *Simoni et al., 2020*). This resistance arises from Cas9/gRNA-directed DNA cuts being repaired by alternative DNA end-joining (EJ) repair pathways, including non-homologous (NHEJ) and microhomology-mediated end-joining (MMEJ), which can introduce insertions or deletions (*indels*) at the target site(s). Many of these *indels* produce loss-of-function (LOF) alleles, which can be selected against if deleterious to the organism. However, functional in-frame EJ-induced *indel* alleles can also be generated, which are unrecognized by the same Cas9/gRNA complex and become drive resistant alleles. When resistant alleles are induced in germ cells, they are heritable and can hinder spread of HGDs (*Champer et al., 2017*; *Hammond et al., 2017*; *Kandul et al., 2020*; *KaramiNejadRanjbar et al., 2018*; *Oberhofer et al., 2018*). Both induced and naturally existing resistant alleles can pose significant challenges to engineering a stable HGD capable of spreading and persisting long term in a population.

To overcome the accumulation of drive resistant alleles, CRISPR-based toxin-antidote (TA) drives, in which embryos are essentially 'poisoned' and only those embryos harboring the TA genetic cassette are rescued, were described (Figure S8 in *Kandul et al., 2019*) and engineered (*Champer et al., 2020a*; *Oberhofer et al., 2020a*, *Oberhofer et al., 2020b*, *Oberhofer et al., 2019*). Generally these designs utilize a toxin consisting of a non-HGD harboring multiple gRNAs targeting a vital gene, and an 'addictive' antidote that is a re-coded, cleavage-immune version of the targeted gene. These TA-based drives are Mendelianly transmitted and spread instead by killing progeny that fail to inherit the drive (e.g. 50% perish from heterozygous mother). Alternative HDR-based TA designs were also described (*Champer et al., 2016*; *Esvelt et al., 2014*), modeled (*Noble et al., 2017*), and recently tested in mosquitoes (*Adolfi et al., 2020*) targeting a recessive non-essential eye pigmentation gene, and in *Drosophila melanogaster* targeting either haplosufficient (i.e. the non-functional allele is recessive as a single functional copy of the target gene is sufficient for normal function) genes (*Terradas et al., 2021*) or a rare haploinsufficient (i.e. the non-functional allele is dominant as a single functional copy of the target gene is not sufficient for normal function) gene (*Champer et al., 2020b*), each demonstrating drive capacity.

Building upon prior work, here we describe the development of a home-and-rescue (HomeR) split-drive (i.e. Cas9 separated from the drive) targeting an essential, haplosufficient gene in *D. melanogaster*. We demonstrate that the accumulation of EJ-induced resistant alleles can be reduced by strategically following four design criteria. First, designing the HGD to target the 3' coding sequence of an essential gene required for insect viability. Second, encoding a dominant rescue of the endogenous target gene into HomeR. Third, using an exogenous 3' UTR to prevent expected deleterious recombination events between the drive and the endogenous target gene. Fourth, by exploiting a process we previously first described as lethal biallelic mosaicism (LBM) in which maternal carryover of Cas9/gRNA complexes contributes to RNA-guided dominant biallelic disruption of an essential target gene throughout development thereby ensuring recessive non-functional resistant alleles result in dominant deleterious/lethal mutations that can get negatively selected out of a population (*Kandul et al., 2019*). Importantly, individuals that inherit the drive allele express a dominant re-coded rescue and are protected from LBM. We demonstrate that efficient cleavage of the target sequence by HomeR and rescue are requisites to achieve nearly ~100% transmission in the presence of Cas9, which is accomplished mostly by homing in trans-heterozygous females. Further, we perform multigenerational population drive experiments demonstrating long-term stability and efficient Cas9-dependent drive. Finally, we conduct comprehensive mathematical modeling to demonstrate that HomeR can outperform contemporary gene drive systems for population modification over wide ranges of fitness, transmission rates, introduction frequencies, and release regimes. Given the simplistic design, this system could be adapted to other species.

## Results

### Selection of *PolG2* as a HomeR drive target

To develop a HomeR-based drive, we first identified an essential haplosufficient gene to target. We chose *DNA Polymerase gamma subunit 2* (*PolG2*, *DNA polymerase γ 35 kDa*, CG33650), required for the replication and repair of mitochondrial DNA (mtDNA; *Carrodeguas and Bogenhagen, 2000*; *Carrodeguas et al., 2001*) whose LOF results in lethality (*Iyengar et al., 2002*). *PolG2* encodes the small subunit of the mtDNA polymerase gamma, acting together with the large subunit 1 (*PolG1, DNA polymerase γ 125 kDa,* CG8987) for the replication and repair of the mitochondrial genome (*Carrodeguas and Bogenhagen, 2000*; *Carrodeguas et al., 2001*). *PolG2* is a short conserved gene with an ~130 amino acids (AA) C-terminal domain (cd02426; *Lu et al., 2020*) sharing ~55% AA identity with the human *PolG2* (*Lecrenier et al., 1997*; *Figure 1—figure supplement 1A–B*). Importantly, *Drosophila PolG2* LOF mutations are known to confer lethality at the early pupal stage (*Iyengar et al., 2002*). The C-terminal location of the functional domain in *PolG2* facilitates minimal re-coding, making *PolG2* an optimal target for a HomeR gene drive (*Figure 1A*, *Figure 1—figure supplement 1A*).

### Assessment of gRNAs targeting *PolG2*

Given that separate gRNAs can result in varying degrees of cleavage efficiencies (*Kandul et al., 2019*), we tested two separate gRNAs targeting the C-terminal domain of *PolG2* (gRNA#1$^{PolG2}$ and gRNA#2$^{PolG2}$) (*Figure 1—figure supplement 1A–B*). According to the *D. melanogaster* Genetic Reference Panel 2 (DGRP2) that includes natural variation in genome architecture among 205 *D. melanogaster* lines (*Huang et al., 2014*; *Mackay et al., 2012*), both gRNA target sequences are completely devoid of any single nucleotide polymorphisms (SNPs) indicating a high degree of conservation. To genetically assess the efficiency of *Cas9/gRNA*-mediated cleavage induced by each gRNA, we crossed these established gRNA lines to two separate Cas9 expressing lines including: (i) a previously characterized ubiquitously expressing Cas9 line (*Port et al., 2014*) in the DNA ligase four null genetic background (*Act5C-Cas9*; *Lig4–/–*) (*Zhang et al., 2014*), and a (ii) germline-enriched Cas9 driven by the *nanos* promoter (*nos-Cas9*) (*Kandul et al., 2020*; *Kandul et al., 2019*; *Figure 1—figure supplement 1CD*). We tested the Cas9/gRNA-mediated cleavage in a *Lig4–/–* background, to decrease the activity of DNA repair by the NHEJ pathway (*McVey et al., 2004*). As the *Lig4* gene is located on the X chromosome, maternal *Lig4–* alleles will be inherited by all male progeny, making them hemizygous *Lig4–* mutants, while females are heterozygous *Lig4–/+*.

We observed that the genetic cross between either gRNA#1$^{PolG2}$ or gRNA#2$^{PolG2}$ homozygous males to *Act5C-Cas9, Lig4 –/–* homozygous females was lethal for all male progeny (*Figure 1—figure supplement 1C*, *Supplementary file 2*). Notably, gRNA#1$^{PolG2}$ also induced lethality in transheterozygous females harboring *Act5C-Cas9* in the *Lig4 +/–* genetic background, suggesting that gRNA#1$^{PolG2}$ is likely more potent. Furthermore, we found that the Cas9 protein deposited by *nos-Cas9/+* females without inheritance of the *nos-Cas9* transgene, referred to as maternal carryover (*Kandul et al., 2019*; *Lin and Potter, 2016*), was sufficient to ensure lethality of the F$_1$ progeny harboring gRNA#1$^{PolG2}$, while gRNA#2$^{PolG2}$ induced lethality only in a fraction of the F$_1$ gRNA#2$^{PolG2}$/*nos-Cas9* trans-heterozygous flies, independent of sex (*Figure 1—figure supplement 1D*, *Supplementary file 3*). Sanger sequencing of trans-heterozygous pupae revealed expected mutations at *PolG2* gRNA target sites. As we previously first described, the mechanism ensuring lethality results from a dominant process we coined LBM (*Kandul et al., 2019*), in which maternal carryover/zygotic expression results in mosaic target gene cleavage throughout development leading to wide scale loss of target gene function which can be detrimental to viability of the organism if essential genes are targeted. Taken together, these results indicate that both tested gRNAs induced cleavage of the *PolG2* target sequences, though gRNA#1$^{PolG2}$ induced greater cleavage than gRNA#2$^{PolG2}$ as evidenced by complete lethality of females and males with both sources of Cas9.

### Development of split HomeR drives with encoded rescue

Using these characterized gRNAs described above (gRNA#1$^{PolG2}$ or gRNA#2$^{PolG2}$), we engineered two *Pol2* HomeR (HomeR$^{PolG2}$ and HomeR(B)$^{PolG2}$) drives, respectively. Fitting with the split-drive (i.e. two-locus) design (*Champer et al., 2020a*; *Kandul et al., 2020*; *Li et al., 2020*), neither

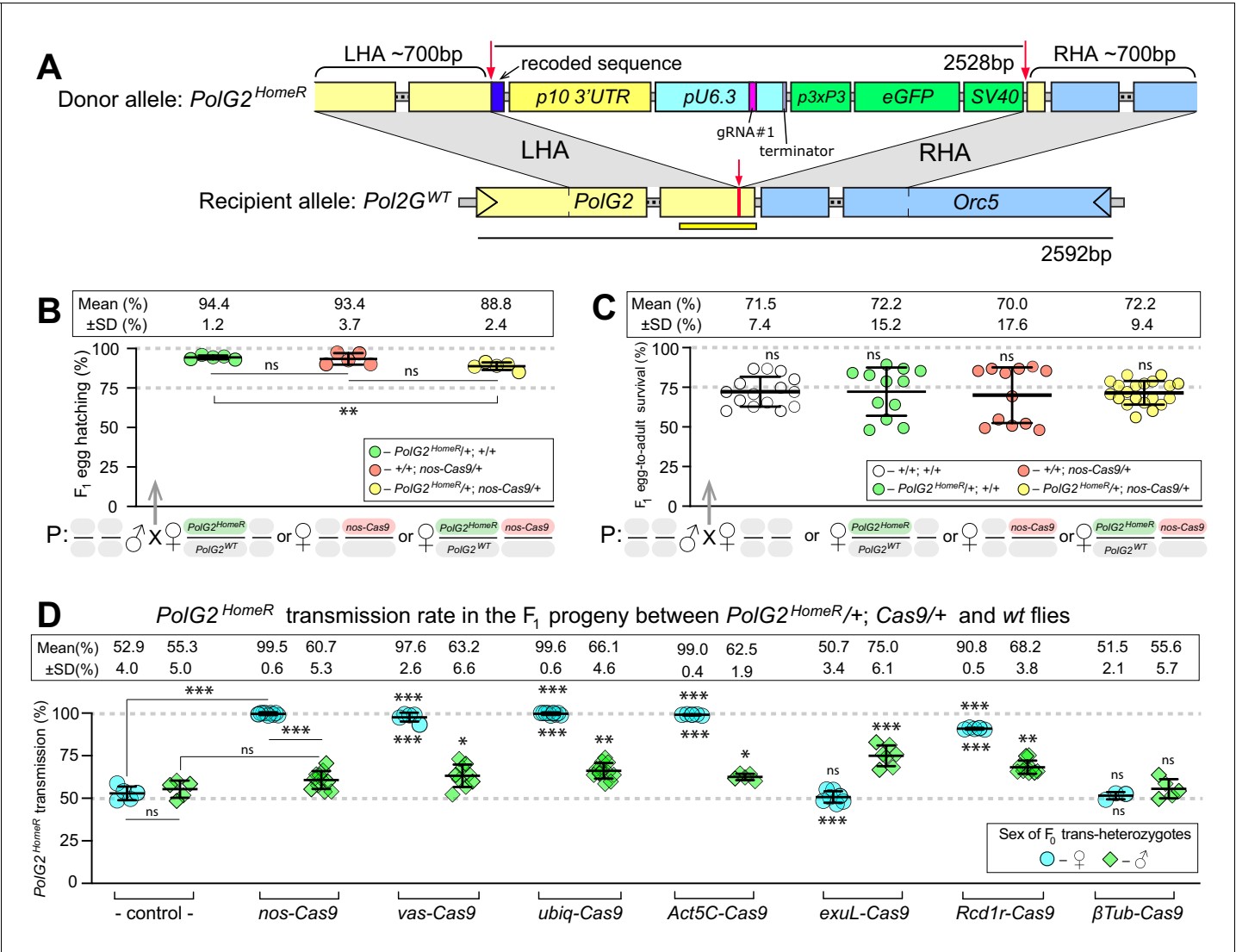

**Figure 1.** *HomeR* homes in the presence of Cas9 and biases its transmission in females to nearly 100%. (**A**) Schematic maps of the *PolG2* donor allele harboring *HomeR* integrated at the *gRNA#1^PolG2* cut site (*PolG2^HomeR*, **Figure 1—figure supplement 1A**), and the recipient wildtype (*wt*) allele (*PolG2^WT*) encompassing the area spanning *PolG2* and *Orc5* (CG7833) genes. To facilitate site-specific integration, the *HomeR* element is surrounded by the left and right homology arms (LHA and RHA) from the Cas9/guideRNA (Cas9/gRNA) cut site (red arrows and lines) in the *wt* allele. The re-coded 3′ end sequence of *PolG2* is shown in dark blue. The yellow line under the *PolG2^WT* recipient allele depicts the location of the C-terminal domain (**Figure 1—figure supplement 1A**). (**B**) Embryonic lethality of *PolG2^WT* alleles cannot result in the nearly 100% transmission of *PolG2^HomeR*. (**C**) The egg-to-adult survival rate indicates that the developmental lethality of *PolG2^WT* alleles also cannot account for the preferential transmission of *PolG2^HomeR*. Therefore, the homing of *PolG2^HomeR* into *PolG2^WT* alleles causes the super-Mendelian transmission of *HomeR*. (**D**) *PolG2^HomeR* supports super-Mendelian transmission in conjunction with different *Cas9* transgenes and/or maternal carryover of Cas9 protein (**Figure 1—figure supplement 3**). Trans-heterozygous females (♀) and males (♂) harboring paternal *Cas9* expressed under different promoters were mated to *wt* flies of the opposite sex, and F$_1$ progeny were scored for the GFP dominant marker of *PolG2^HomeR*. The transmission rate was compared to that in *PolG2^HomeR*/*PolG2^WT*; +/+ flies without *Cas9* (control) of the corresponding sex (statistical significance indicated above data points). In addition, the transmission rate by trans-heterozygous females was compared to that of trans-heterozygous males for each *Cas9* promoter (statistical significance indicated below data points). Notably, while *PolG2^HomeR* can bias its transmission in both sexes, the highest transmission rate is achieved in *Drosophila* females: 99.6 ± 0.6% in ♀ vs. 75.0 ± 6.1% in ♂. Plots show the mean ± SD over at least five biological replicates. Statistical significance was estimated using a two-sided Student's *t* test with equal variance (p ≥ 0.05^ns, p<0.05*, p<0.01**, and p<0.001***).

The online version of this article includes the following figure supplement(s) for figure 1:

**Figure supplement 1.** Assessing gRNAs for Cas9-mediated cleavage of *PolG2*.

**Figure supplement 2.** Site-specific integration of *HomeR* constructs at the *PolG2* locus.

**Figure supplement 3.** The gRNA efficiency to guide Cas9-mediated cleavage limits the transmission rate for *PolG2^HomeR*.

**Figure supplement 4.** Mechanism of lethal biallelic mosaicism (LBM).

*HomeR*[PolG2] nor *HomeR(B)*[PolG2] include the *Cas9* gene and thus are inherently confinable drives (*Akbari et al., 2015*; *Champer et al., 2016*; *Esvelt et al., 2014*; *Marshall and Akbari, 2018*). To mediate HDR, both HomeR constructs include left and right homology arms (LHA and RHA) matching the sequences surrounding the cut site of the corresponding gRNA. The LHA includes a carefully re-coded sequence of 22 or 27 AA downstream from the cut site #1 or #2 (*Figure 1—figure supplement 1A–B*), and a p10 3' UTR to support robust expression of the re-coded *PolG2* and to prevent gene conversion between the rescue HomeR allele and the endogenous allele, which proved problematic in previous drive design architectures (*Champer et al., 2020a*; *Champer et al., 2020b*). Additionally, we included a dominant *3xP3-eGFP-SV40* marker gene to visually track the presence of HomeR (*Figure 1—figure supplement 2*).

Two different approaches were used to generate transgenic lines harboring site-specific integrations of *HomeR*[PolG2] and *HomeR(B)*[PolG2] constructs at the corresponding homing cut sites in *PolG2* (termed *PolG2*[HomeR] and *PolG2*[HomeR(B)] when integrated into genome). In the first approach, the constructs were initially randomly integrated into the genome and then relocated precisely into the *PolG2* cut sites via Homology Assisted CRISPR Knock-in (HACK; *Gantz and Akbari, 2018*; *Lin and Potter, 2016*). In the second approach, the constructs were directly integrated into the *PolG2* cut sites by injecting them into *nos-Cas9* embryos (*Kandul et al., 2019*; *Figure 1—figure supplement 2B*). Using both approaches, multiple independent transgenic lines of each *PolG2*[HomeR] and *PolG2*[HomeR(B)] were generated. To confirm that *PolG2*[HomeR] or *PolG2*[HomeR(B)] lines were indeed inserted precisely at the corresponding target site in *PolG2*, we assessed their ability for super-Mendelian inheritance in the presence of *Cas9* in trans. Establishment of pure breeding, viable homozygous stocks of *PolG2*[HomeR]/*PolG2*[HomeR] and *PolG2*[HomeR(B)]/*PolG2*[HomeR(B)], demonstrated a functional rescue of *wt PolG2* function. Finally, we Sanger-sequenced the junction sites (*Figure 1—figure supplement 2C*) and molecularly confirmed the precision of HDR-mediated insertions.

## Assessment of germline transmission and cleavage rates

To assess the effect of gRNA-mediated cleavage efficiency on the inheritance of *HomeR*, we compared the two *HomeRs*, as they harbored two distinct gRNA sequences that differed in cleavage efficiencies. The *PolG2*[HomeR] and *PolG2*[HomeR(B)] lines encode *gRNA#1*[PolG2] and *RNA#2*[PolG2], respectively, with slightly different LHA and RHA corresponding to their respective gRNA cut sites (*Figure 1—figure supplement 2A*). We found that *PolG2*[HomeR]/+; *nos-Cas9*/+ trans-heterozygous females crossed to *wt* males transmitted *PolG2*[HomeR] to 99.5 ± 0.6% of progeny, while *PolG2*[HomeR(B)]/+; *nos-Cas9*/+ females transmitted the corresponding *PolG2*[HomeR(B)] to a significantly lower fraction of $F_1$ progeny (68.7 ± 6.2%, two-sided Student's *t* test with equal variances, p<0.0001; *Figure 1—figure supplement 3*). Genetic crosses of either *PolG2*[HomeR]/+; *nos-Cas9*/+ or *PolG2*[HomeR(B)]/+; *nos-Cas9*/+ trans-heterozygous males to *wt* females did not result in significant biased transmission to $F_1$ progeny (60.7 ± 5.3% vs. 52.9 ± 4.0% or 54.3 ± 4.0% vs. 51.5 ± 1.8%, respectively, two-sided Student's *t* test with equal variances, p>0.05; *Figure 1—figure supplement 3*, *Supplementary file 4–5*). Maternal carryover of Cas9 protein by *nos-Cas9*/+ females significantly increased transmission of *PolG2*[HomeR] by $F_1$ *PolG2*[HomeR]/*CyO* females, 66.1 ± 0.8% vs. 52.9 ± 4.0% (two-sided Student's *t* test with equal variances, p<0.001; *Figure 1—figure supplement 3*, *Supplementary file 4*). These results suggest that the higher efficiently of *gRNA#1*[PolG2] to guide the Cas9-mediated *PolG2* disruption, which results in lethality (*Figure 1—figure supplement 1C–D*), likely contributes to the higher transmission rates of *PolG2*[HomeR], and underscores the importance of selecting an efficient gRNA for engineering gene drives.

## HomeR biases its inheritance predominantly by homing

We hypothesized that either homing (indicating allelic conversion) in germ cells (*Figure 1A*) or 'destruction' of *wt* alleles in the progeny of trans-heterozygous *PolG2*[HomeR]/+; *Cas9*/+ females via LBM (*Figure 1—figure supplement 4*; *Kandul et al., 2019*) could contribute to biased *PolG2*[HomeR] transmission rates. LBM contributes to dominant biallelic disruption of the target gene throughout development thereby ensuring recessive non-functional resistant alleles (R2 type) result in dominant deleterious/lethal mutations that can get selected out of a population (*Figure 1—figure supplement 4*). Previously, destruction of the *wt* allele in conjunction with maternal carryover of a 'toxin' was used to engineer gene drives based on an 'addictive' TA approach (*Champer et al., 2020a*;

*Oberhofer et al., 2020a*, *Oberhofer et al., 2019*). In these TA drives, one half of the $F_1$ progeny did not inherit the TA cassette, that is, not rescued, and were killed—ensuring survival of only progeny inheriting the drive resulting in a rapid spread of the genetic cassette throughout laboratory populations.

To explore the mechanism resulting in the super-Mendelian inheritance of $PolG2^{HomeR}$, we determined the egg hatching and egg-to-adult survival rates for the progeny of trans-heterozygous females and compared it to those of females heterozygous for $PolG2^{HomeR}$ or just $Cas9$ (*Figure 1B, C*). The hatching rate of $F_1$ eggs generated by $PolG2^{HomeR}/PolG2^{WT}$; *nos-Cas9*/+ trans-heterozygous females crossed to *wt* males was reduced by 5.6% as compared to that of $PolG2^{HomeR}/PolG2^{WT}$; +/+ heterozygous females (88.8 ± 2.4% vs. 94.4 ± 1.2%; two-sided Student's *t* test with equal variances, p<0.004, *Figure 1B*) and slightly lower than +/+; *nos-Cas9*/+ heterozygous females (88.8 ± 2.4% vs. 93.4 ± 3.7%; two-sided Student's *t* test with equal variances, p=0.052; *Figure 1B*, *Supplementary file 6*), suggesting some degree of embryo killing. Furthermore, we observed no significant difference among egg-to-adult survival rates estimated for four female types crossed to *wt* males: 72.2 ± 9.4% for $PolG2^{HomeR}/PolG2^{WT}$; *nos-Cas9*/+ ♀, 72.2 ± 15.2% for $PolG2^{HomeR}$/+ ♀, 70.0 ± 17.6% for *nos-Cas9*/+ ♀, and 71.5 ± 7.4% for *wt* ♀ (*Figure 1C*, *Supplementary file 7*). Taken together, these data indicate that only a small fraction of $PolG2^{WT}$ alleles transmitted by trans-heterozygous females were 'destroyed' via LBM—meaning mutated and not complemented by the paternal $PolG2^{WT}$ allele, since it was also mutated by Cas9/gRNA maternal carryover (*Figure 1—figure supplement 4*). Therefore, the *HomeR* transmission of 99.5% by the $PolG2^{HomeR}/PolG2^{WT}$; *nos-Cas9*/+ females could not be explained simply by the 'destruction' of $PolG2^{WT}$ alleles, which would result in the lethality of 50% progeny as in cleave and rescue (ClvR; *Oberhofer et al., 2020a*; *Oberhofer et al., 2020b*; *Oberhofer et al., 2019*) and toxin-antidote recessive embryo (TARE; *Champer et al., 2020a*) drives. Instead, *HomeR* biases its transmission predominantly by homing (i.e. allelic conversion of $PolG2^{WT}$ into $PolG2^{HomeR}$) from trans-heterozygous females (*Figure 1A*).

## HomeR exhibits the strong transmission bias from females

The split-drive design facilitates testing of different Cas9 promoters. Therefore, we quantified the transmission of $PolG2^{HomeR}$ from either trans-heterozygous females or males harboring $PolG2^{HomeR}$ in combination with four Cas9 promoters active in germ cells of both sexes (*Figure 1D*). *Nanos* (*nos*) and *vasa* (*vas*) promoters were previously described as germline-specific promoters active in both sexes (*Hay et al., 1988*; *Sano et al., 2002*; *Van Doren et al., 1998*), though recent evidence indicates ectopic expression in somatic tissues from both *nos-Cas9* and *vas-Cas9* (*Kandul et al., 2020*; *Kandul et al., 2019*). The *Ubiquitin 63E (ubiq)* and *Actin 5C (Act5C)* promoters support strong expression in both somatic and germ cells (*Kandul et al., 2020*; *Kandul et al., 2019*; *Port et al., 2014*; *Preston et al., 2006*). Since maternal carryover of the Cas9 protein was shown to induce a 'shadow drive' two generations later (*Guichard et al., 2019*; *Kandul et al., 2020*), we used trans-heterozygous flies that inherited paternal *Cas9* to quantify the transmission of $PolG2^{HomeR}$. Trans-heterozygous females harboring $PolG2^{HomeR}$ together with *nos-Cas9*, *vas-Cas9*, *ubiq-Cas9*, or *Act5C-Cas9* crossed to *wt* males biased transmission of $PolG2^{HomeR}$ to nearly ~100% of $F_1$ progeny (99.5 ± 0.6%, 97.6 ± 2.6%, 99.6 ± 0.6%, and 99.0 ± 0.4%, respectively, vs. 52.9 ± 4.0% by $PolG2^{HomeR}/PolG2^{WT}$; +/+ females, two-sided Student's *t* test with equal variances, p<0.001; *Figure 1D*). Note that the corresponding trans-heterozygous males only modestly biased $PolG2^{HomeR}$ transmission from 55.3 ± 5.0% of $F_1$ progeny to 60.7 ± 5.3% (p>0.05), 63.2 ± 6.6% (p<0.03), 66.1 ± 4.6% (p<0.004), and 62.0 ± 1.7% (p<0.017, two-sided Student's *t* test with equal variances, *Figure 1D*, *Supplementary file 4*), respectively.

To assess whether males could support robust homing, we investigated three alternative male-specific promoters. We established the *Drosophila exuperantia* (CG8994) large fragment (*exuL*) promoter for an early male-specific expression. The *Rcd-1 related* (*Rcd1r*, CG9573; *Chan et al., 2013*) and *βTubulin 85D* (*βTub*; *Chan et al., 2011*; *Michiels et al., 1989*) promoters support an early and late, respectively, testis-specific expression in *Drosophila* males. We found that only *exuL-Cas9* induced the male-specific super-Mendelian inheritance of $PolG2^{HomeR}$; trans-heterozygous males, but not females, transmitted $PolG2^{HomeR}$ to more than 50% of $F_1$ progeny (75.0 ± 6.1% vs. 55.3 ± 5.0% in ♂, p<0.0001; and 50.7 ± 3.4% vs. 52.8 ± 4.0% in ♀, p>0.05, two-sided Student's *t* test with equal variances; *Figure 1D*, *Supplementary file 4*). To our surprise, *Rcd1r-Cas9* induced super-

Mendelian inheritance of $PolG2^{HomeR}$ in both trans-heterozygous males and females (68.2 ± 3.8% vs. 55.3 ± 5.0% in ♂, p<0.002; and 90.8 ± 0.5% vs. 52.8 ± 4.0% in ♀, p>0.0001, two-sided Student's $t$ test with equal variances; *Figure 1D*). Finally, $βTub$-$Cas9$ did not induce changes in transmission of $PolG2^{HomeR}$ by either trans-heterozygous males or females (55.6 ± 5.7% vs. 55.3 ± 5.0% in ♂, p=0.55; and 51.5 ± 2.1% vs. 52.9 ± 4.0% in ♀, p=0.94, two-sided Student's $t$ test with equal variances; *Figure 1D*, *Supplementary file 4*). These results suggest that *Drosophila* males bias $PolG2^{HomeR}$ transmission; however, this bias is substantially lower than the nearly ~100% transmission of $PolG2^{HomeR}$ in females.

## Induced resistant alleles do not impede drive invasion

We reasoned that insertion of HomeR into an essential gene could enable spread into a population by biasing transmission while also inhibiting the accumulation of LOF resistant alleles (R2 type, $PolG2^{R2}$) through a combination of slow-acting Mendelian selection and by LBM (*Figure 1—figure supplement 4*). However, functional resistant alleles (R1) that alter the AA sequence of the target can indeed still be generated (e.g. by EJ repair resulting from in-frame *indels,* or nonsynonymous base substitutions) and may hinder drive spread. While these mutations may partially rescue the function of the target gene resulting in viability, it has not escaped our attention that they may still confer detrimental fitness costs to their carriers resulting from incompletely preserving the function of the essential target gene due to the altered AA sequence(s) in a critical domain and may result in negative selection within a population. We therefore refer to these here as 'non-silent R1' or 'non-silent $PolG^{R1}$' mutations. On the contrary, resistant alleles that alter the nucleotide sequence, but not the AA sequence, can also be generated, and these are referred to here as 'silent R1' or 'silent $PolG^{R1}$ mutations'. These silent R1 mutations are expected to be especially problematic to drive spread as these would not be predicted to impose fitness costs on homozygous carriers, and therefore they would be expected to spread at the expense of the drive. Notwithstanding, the generation of both types of R1 functional resistant alleles (non-silent/silent) is a shared problem of all HGDs developed to date.

To explore this potential, in addition to spread and stability, we initiated three multigenerational cage populations of heterozygous $PolG2^{HomeR}$/+ flies (50% allelic frequency) in the *nos-Cas9/nos-Cas9* genetic background and assessed if induced resistant alleles could impede drive invasion and stability in 10 discrete generations (*Figure 2A*). Functional resistant alleles (R1) were expected to be generated, especially at the earlier generations initiated with 50% *wt* alleles, and would be straightforward to score in our assay by a simple loss of the dominant GFP marker. Note that any viable GFP-negative (GFP–) adult flies must have at least one functional $PolG2$ allele to survive (either $PolG2^{WT}$ or $PolG2^{R1}$).

In early generations 2 and 3, we sampled nine such flies lacking $PolG2^{HomeR}$ in two out of three populations (#1 and #3; *Figure 2A*, *Supplementary file 8*). To ensure that the $PolG2^{R1}$ alleles had a chance to transmit and compete with $PolG2^{HomeR}$ alleles, these flies were transferred among the subsequent generation and allowed to mate with other flies and lay eggs in each population lineage before genotyping. As expected, we determined that each fly indeed harbored at least one $PolG2^{R1}$ allele that rescued viability. Two distinct non-silent $PolG2^{R1}$ alleles were identified, with one non-silent $PolG2^{R1}$ type induced independently in two populations (*Figure 2—figure supplement 1A*). Since we did not find any fly without the $PolG2^{HomeR}$ allele after generation 3, it can be noted that the identified non-silent $PolG2^{R1}$ resistant alleles did not impede drive invasion/stability. Despite this, we cannot rule out the possibility that these and other resistant alleles were still present at low frequencies in populations masked by $PolG2^{HomeR}$ alleles.

To further explore the diversity of resistant alleles remaining after 10 generations, we performed next-gen sequencing on 60 randomly selected GFP+ flies (note that each fly had at least one copy of $PolG2^{HomeR}$) from each population to identify and quantify mutant $PolG2$ alleles, which did not harbor the large insert of $HomeR^{PolG2}$ (~2.5 kb; *Figure 1A*). From nearly 150,000 sequence reads generated, we did not identify the two previously sampled functional non-silent $PolG2^{R1}$ alleles (*Figure 3—figure supplement 1A*). Instead, we found two novel non-silent $PolG2^{R1}$ in-frame *indel* alleles, 18 and 9 bp deletions, in populations #2 and #3 (*Figure 2—figure supplement 1B*). Additionally, we found 11 LOF $PolG2^{R2}$ alleles harboring out-of-frame *indels* ranging from a 1 bp insertion to a 23 bp deletion (*Figure 2—figure supplement 1B*). Two $PolG2^{R2}$ alleles, 2 and 4 bp deletions, were also seen in the genotyped flies at generations 2 and 3, suggesting that these may

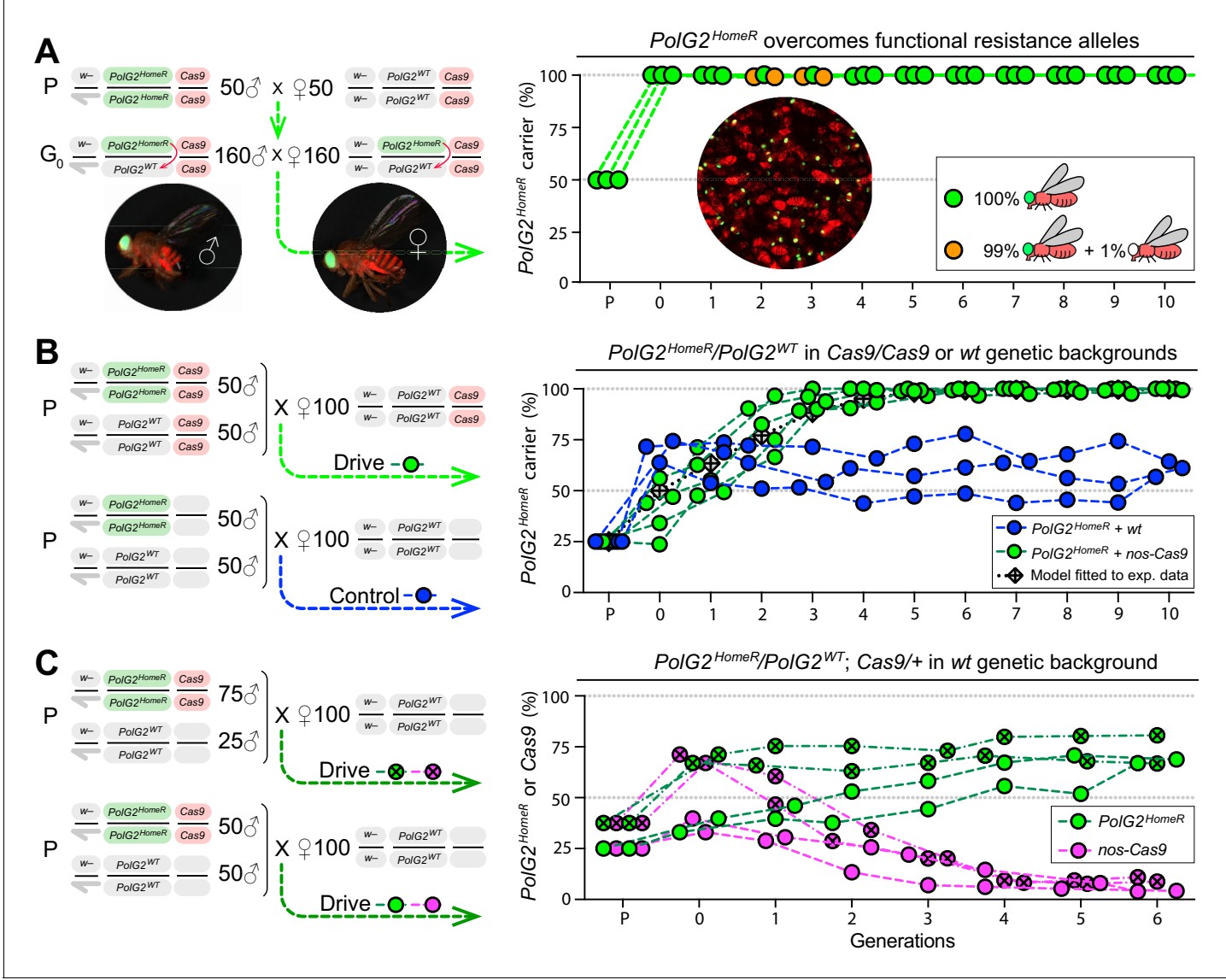

**Figure 2.** Induced resistant alleles do not impede the Cas9-dependent spread of *PolG2^HomeR^*. (A) To explore the fate of induced functional resistant alleles (R1), three population cages were seeded with *PolG2^HomeR^*/*PolG2^WT^* heterozygous flies in *nos-Cas9*/*nos-Cas9* genetic background and run for 10 discrete generations. *PolG2^HomeR^* and *nos-Cas9* were tagged by dominant eye-specific GFP and body-specific dsRed, respectively. Images of an individual male (♂), female (♀), and a group of flies are shown. In total, nine *PolG2^HomeR^*-negative flies, as determined by the absence of eye-specific GFP, were identified at generations 2 and 3 in populations #1 and #3. After these flies participated in seeding the next generation, they were isolated and genotyped. Two functional *PolG2^R1^* resistant alleles identified in these flies (*Figure 2—figure supplement 1A*) were not sampled in flies collected at generation 10 by Illumina amplicon sequencing (*Figure 2—figure supplement 1B*). (B) The *PolG2^HomeR^* allele spread efficiently in the homozygous *Cas9* genetic background. Population drives were seeded with 50 *PolG2^HomeR^*/*PolG2^HomeR^* ♂, 50 *wt* ♂, and 100 *wt* virgin ♀ in the presence (green points) or absence (blue points) of *nos-Cas9*, and the carrier frequency of *PolG2^HomeR^* was scored at each discrete generation. The model for the HomeR population replacement drive (gray diamonds and a black dotted line) was fitted to the empirical data of the *PolG2^HomeR^* spread in the presence of *nos-Cas9* (green points). After 10 generations, the *PolG2^HomeR^* allele spread from the introductory frequency of 25% to the carrier frequency of 99.9 ± 0.3% in the presence of *nos-Cas* (green points) or continued to drift at 60.8 ± 3.8% without the *Cas9* transgene (blue points; p<0.0001, a two-sided Student's *t* test with equal variance). (C) The invasion of *PolG2^HomeR^* is limited by the fitness of the *Cas9* transgene. Double homozygous males at frequencies of 75% or 50% were released into the *wt* genetic background to establish four drive populations with mixtures of trans-heterozygous and *wt* flies at above or below 50%, respectively, in generation 0. Both *PolG2^HomeR^* and *nos-Cas9* transgenes were scored at each generation by the GFP (green points) and dsRed (purple points) markers. The carrier frequency of *nos-Cas9* decreased from 69.1% to 9.9% or from 36.4% to 4.0% in six generations confining the spread of *PolG2^HomeR^*.

The online version of this article includes the following figure supplement(s) for figure 2:

**Figure supplement 1.** *PolG2* resistant alleles induced by *PolG2^HomeR^* in experimental populations.

have persisted in the populations. The relative abundance of each allele can be used to extrapolate the minimum number of resistant alleles sampled in the 60 heterozygous and/or homozygous flies. We inferred that at least 9, 5, and 17 resistant alleles persisted for 10 generations and were rescued by the *PolG2^HomeR^* allele in 60 sampled flies from populations #1, #2, and #3, respectively (*Figure 2—figure supplement 1B*). Since a single *PolG2^HomeR^* allele rescues the *wt* function of *PolG2* and can mask the opposite allele at the *PolG2* locus from slow-acting purifying selection, it is not surprising that LOF resistant alleles can be found persisting in the population.

## HomeR spreads in a Cas9-dependent manner

To evaluate drive efficacy in the presence of Cas9, we established five drive and three control ('no-drive') populations by seeding 50 homozygous *PolG2^HomeR^* males and 50 *wt* males together with 100 *wt* virgin females in the presence or absence of *Cas9,* respectively (*Figure 2B*). The introduction ratio of *PolG2^HomeR^* to *PolG2^WT^* was 1:2 (or 25% allele frequency) in the parental generation (P). Both types of homozygous *PolG2^HomeR^* males with and without *Cas9* were able to compete with the corresponding *wt* males and sired 41.0 ± 12.4% (*Supplementary file 9*) and 70.0 ± 5.5% (*Supplementary file 10*) of progeny, respectively. Notably, the *PolG2^HomeR^* males were significantly less competitive with *wt* males for female mates in the *nos-Cas9* genetic background than in the *wt* genetic background (p=0.01, two-sided Student's *t* test with equal variances; *Figure 2B*). Nevertheless, the *PolG2^HomeR^* allele spread to the carrier frequency of 96.7 ± 4.4% in the presence of *nos-Cas9* vs. 56.9 ± 11.6% without the *Cas9* transgene in a time span of four generations (p=0.0004, a two-sided Student's *t* test with equal variance; *Supplementary file 9–10*). At generation 10, the *PolG2^HomeR^* allele was fixed in four out of five drive populations and continued to drift at moderate frequency in three control populations in the absence of *Cas9*: 99.9 ± 0.3% vs. 60.8 ± 3.8%, respectively (p<0.0001, a two-sided Student's *t* test with equal variance; *Figure 2B*, *Supplementary file 9–10*), underscoring Cas9 dependence for drive.

A few GFP– flies harboring *PolG2^R1^* alleles appeared over multiple generations in drive populations #4 and #5 (*Supplementary file 9*). To assess the fertility of viable *PolG2^R1^* carriers, we collected 7 GFP– females and 7 GFP– males at generation 9 and individually crossed them to *wt* flies of the corresponding sex. Interestingly, we found that the GFP– males were fertile, while each tested GFP– female died in 3 days without producing progeny suggesting that the sampled *PolG2^R1^* allele(s) incurred fitness costs to female carriers. Dead females were genotyped and we identified four non-silent R1 alleles that rescued their 'short-lived' viability (*Figure 2—figure supplement 1C*). Notably one non-silent R1 allele was already sampled in the GFP– flies from heterozygous population #1 (R1#2 in *Figure 2—figure supplement 1C*). Each tested GFP– male was fertile, and four genotype males had the same non-silent allele identified in the females (R1#3 in *Figure 2—figure supplement 1C*). In summary, all in-frame resistant alleles identified in this study resulted in AA changes and are non-silent and importantly we did not sample any silent *PolG2^R1^* mutant alleles.

Fitting a mathematical model of CRISPR/Cas9-based homing drive to the observed cage data in *Figure 2B* (see 'Materials and methods'), we found the data to be consistent with cleavage efficiencies in females and males of 99.2% (95% credible interval [CrI] 96.4–100%) and 99.6% (95% CrI 98.2–100%), respectively, and a frequency of accurate HDR, given cleavage, in females and males of 99.5% (95% CrI 97.8–100%) and 9.6% (95% CrI 8.0–10.0%), respectively. When accurate HDR did not occur, the data were consistent with 2.9% (95% CrI 1.9–4.0%) of resistant alleles being in-frame, and the remainder being out-of-frame or otherwise costly LOF alleles. Individuals having the *HomeR* system were found to have a negligible fitness cost of 0.3% (95% CrI 0.0–1.4%), while individuals homozygous for the LOF allele were modeled as completely unviable. The fitted parameter estimates are consistent with parameters estimated from individual pair crossings (*Figure 1D*).

## The spread of HomeR is confined by the fitness of *Cas9*

In the split-drive (i.e. two-locus) design, the continued spread of *HomeR* is contingent on the availability of *Cas9* which ensures confinability. Therefore, to explore the invasion potential of *HomeR* under a limited supply of *Cas9*, we seeded additional drive populations with double homozygous and *wt* flies, and scored both *PolG2^HomeR^* and *Cas9* at each generation. Four populations with *PolG2^HomeR^/PolG2^HomeR^*; *nos-Cas9/nos-cas9* and *wt* (*PolG2^WT^/PolG2^WT^*; +/+) males mixed at 1:1 (two replicates; 25% allele frequency) or 3:1 ratios (two replicates; 37.5% allele frequency) and 100

*wt* virgin females were seeded (*Figure 2C*). These ratios generated two population types with frequencies of trans-heterozygous flies below and above 50% in the subsequent generation: 36.4 ± 4.8% and 69.1 ± 3.0% (generation 0 in *Figure 2C*, *Supplementary file 11*). After tracking these populations for six generations, we found that frequencies of the *PolG2$^{HomeR}$* allele gradually increased to 67.9 ± 1.3% and 73.7 ± 9.8%, respectively, and persisted near these frequencies. Notably, we also observed that the frequency of *nos-Cas9* decreased each generation down to 4.1 ± 0.2% or 9.9 ± 1.6% in two population types by generation 6 (*Figure 2C*, *Supplementary file 11*). These results suggest that the *nos-Cas9* incurred fitness costs on its carrier and was therefore negatively selected out from the population. Furthermore, these experiments underscore the significant fitness cost *Cas9* can impose, and indicate that a single release, at the introduction thresholds used here, would be insufficient to achieve fixation of *PolG2$^{HomeR}$* given such immense costs to Cas9. We therefore use mathematical modeling to explore multi-release scenarios (below).

## Modeling indicates that *HomeR* is an efficacious gene drive

To compare the performance of HomeR against contemporary gene drive systems for population modification, we modeled one- (i.e. autonomous, linked-Cas9) and two-locus (i.e. split-drive) versions of ClvR (*Faber et al., 2020*; *Oberhofer et al., 2020a*, *Oberhofer et al., 2020b*, *Oberhofer et al., 2019*), the one-locus TARE system from *Champer et al., 2020a*, as well as a two-locus TARE configuration based on their design, an HGD targeting a non-essential gene (*Gantz et al., 2015*; *Hammond et al., 2016*), and HomeR (for mechanistic comparisons of these systems, see *Figure 3—figure supplement 1*). In each case, we first simulated population spread of each gene drive system for an ideal parameterization (see 'Materials and methods' for more details) and included additional simulations for HomeR under current experimentally derived parameters (HomeR-exp; *Figure 3A and C*, parameters consistent with *Figure 2C*). To gauge behavior across a range of scenarios, we performed simulations for a range of fitness costs (implemented as female fecundity reduction) and drive system transmission rates (implemented by varying the cleavage rate), providing heatmaps of the expected performance for each drive system at each parameter combination (*Figure 3B and D*). Drive efficacy, the outcome in these comparisons, is defined as the expected fraction of individuals that carry the effector allele, in either heterozygous or homozygous form, at 20 generations following a 25% release of male homozygotes for each drive system.

When one-locus gene drive (GD) systems are compared for ideal parameter values, HomeR outperforms all other GDs in terms of speed of spread, and reaches near fixation in terms of carrier frequency, as do ClvR and TARE (*Figure 3A*). HGD displays a similar speed of spread to HomeR initially; however, fitness costs from the targeted gene disruption and LOF (R2) alleles slow the introgression and allow functional resistance alleles (R1) to build up over time, preventing fixation. The HomeR design overcomes this fitness reduction and R2 allele build-up by rescuing the *wt* function of a targeted essential gene (*Figure 3—figure supplement 1*). ClvR and TARE perform similarly to each other for ideal parameter values, but reach near carrier fixation ~4 generations after HomeR does for ideal parameter values (*Figure 3A*). When experimental parameters are used for HomeR (HomeR-Exp, in *Figure 3A*), it reaches near carrier fixation a generation after ClvR; almost on-par with ideal ClvR and TARE systems and significantly better than HGD. HomeR also reaches near carrier fixation for the widest range of fitness and transmission rate parameter values (*Figure 3B*). As an HGD, HomeR drives to high carrier frequencies provided its inheritance bias (or transmission rate) exceeds its associated fitness cost. In contrast, drive efficacy of ClvR and TARE is strongly dependent on fitness cost and weakly dependent on transmission rate. Indeed, ClvR and TARE can each only tolerate fitness costs less than ~20% (*Figure 3B*). This is a consequence of their design, employing a TA scheme, which induces a significant fecundity reduction (*Figure 3—figure supplement 1A–B*) in addition to other fitness costs. A one-locus HGD also exhibits efficacy across a wide range of parameter combinations, but its efficacy is reduced compared to HomeR due to the build-up of R2 and R1 alleles (*Figure 3—figure supplement 1C–D*), which can potentially block spread of the HGD in large populations (*Figure 3B*).

In two-locus simulations, Cas9 is separated from drive in all designs and undergoes independent assortment during gametogenesis. The effects of this design change are evident (*Figure 3C*). Under the same experimental conditions as one-locus simulations, there is significantly more variation in behavior of two-locus GDs, with a reduced speed of introgression into the population and slightly reduced overall efficacy. Nevertheless, general trends remain the same; HomeR with ideal

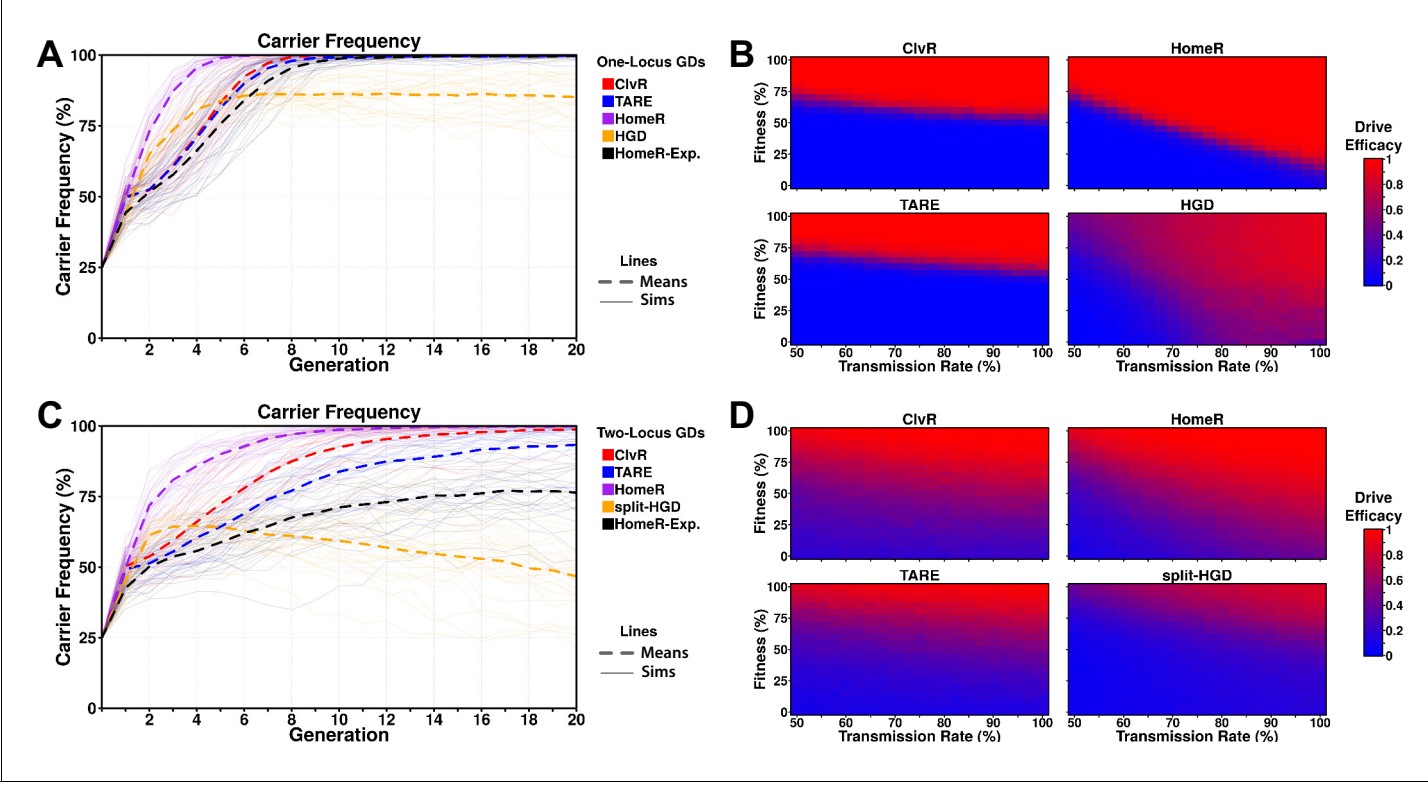

**Figure 3.** Performance of contemporary gene drive systems for population modification with a single release. (**A**) Simulations of carrier frequency trajectories (i.e. heterozygotes and homozygotes) for one-locus versions of ClvR, TARE, HomeR, and HGD for ideal parameters (see 'Materials and methods'), and HomeR for experimental parameters (HomeR-Exp, see 'Materials and methods'). Twenty-five repetitions (lighter lines) were used to calculate the average behavior of each drive (thicker, dashed lines). Populations were initialized with 50% wildtype (*+/+*) adult females, 25% wildtype (*+/+*) adult males, and 25% drive homozygous (*drive/drive*) males. (**B**) Heatmaps depicting drive efficacy for one-locus versions of ClvR, TARE, HomeR, and HGD for a range of fitness and transmission rate parameter values. Fitness costs were incorporated as a dominant, female-specific fecundity reduction. Transmission rate was varied based on cleavage rate, using HDR rates consistent with ideal parameters, when applicable (see 'Materials and methods'). Drive efficacy is defined as the average carrier frequency at generation 20 (approximately 1 year, given a generation period of 2-3 weeks) based on 100 stochastic simulations with the same initial conditions as (**A**). (**C**) Simulations of carrier frequency trajectories for two-locus (split-drive) versions of ClvR, TARE, HomeR, and HGD for ideal parameters (see 'Materials and methods'), and HomeR for experimental parameters (HomeR-Exp, see 'Materials and methods'). Twenty-five repetitions (lighter lines) were used to calculate the average behavior of each drive (thicker, dashed lines). Populations were initialized with 50% wildtype (*+/+; +/+*) adult females, 25% wildtype (*+/+; +/+*) males, and 25% drive homozygous (*Cas9/Cas9; gRNA/gRNA*) males. (**D**) Heatmaps depicting drive efficacy for two-locus versions of ClvR, TARE, HomeR, and HGD for a range of fitness and transmission rate parameter values, implemented as in panel (**B**), with initial conditions given in (**C**).

The online version of this article includes the following figure supplement(s) for figure 3:

**Figure supplement 1.** Mechanistic comparison of contemporary split-drives for population modification.

parameters is more capable than comparable drives, though current experimental realizations require improvement. TARE performs significantly worse in a split configuration (*Champer et al., 2020a*). ClvR, when completely unlinked, also performs significantly worse, in agreement with results from *Oberhofer et al., 2020a*. Exploring the performance under a range of parameters, we found reduced overall efficacy for all drives (*Figure 3D*), but an increased range of lower efficacy for ClvR, HomeR, and TARE. This is consistent with fitness costs applied to the Cas9 locus, which is now separated from the effector gene and gRNAs. HomeR demonstrates the widest range of achieving efficacy as well as the widest range of high expected efficacy.

## Exploration of multi-release scenarios

To probe the ability of these drive designs to modify field populations, we implemented an overlapping generation model (*Sanchez et al., 2019*), performing weekly male releases into a naive population, and tested if the frequency of females carrying the effector allele reached 95% of the female

population, and how long that carrier frequency remained above 95%. One-locus constructs of ClvR, TARE, and HomeR were consistently able to reach this threshold, though HomeR achieved these thresholds over the widest range of transmission rates and fitness costs (*Figure 4A*). HGD never reached this threshold because of R2 allele build-up. This does not indicate that HGD cannot be effective at lower thresholds (indeed, during testing it was), but that even low rates of resistance generation are problematic. HomeR was the only one to consistently remain above a 95% carrier frequency for over 100 days (*Figure 4B*).

Two-locus designs showed significantly reduced ability to reach 95% carrier frequency in females, often requiring more and larger (20% of the total population size) releases to be effective (*Figure 4C*). For split-drive designs, only the first release was homozygous Cas9 and gene drive, while supplementary releases were homozygous Cas9 only (*Faber et al., 2020*; *Oberhofer et al., 2020a*, *Oberhofer et al., 2020b*, *Oberhofer et al., 2019*). A similar pattern of efficacy is seen for ClvR, TARE, and HomeR, but by splitting the HGD and maintaining the fitness effects on the Cas9 allele, it is now able to reach 95% introgression over a small parameter range. Additionally, as fitness costs are predominantly associated with the Cas9 allele and not the effector gene, all constructs were adequate at maintaining effector allele frequency in the population over a long period of time (*Figure 4D*). Taken together, these results suggest that multi-releases would be sufficient to ensure HomeR spreads and persists stably in a population.

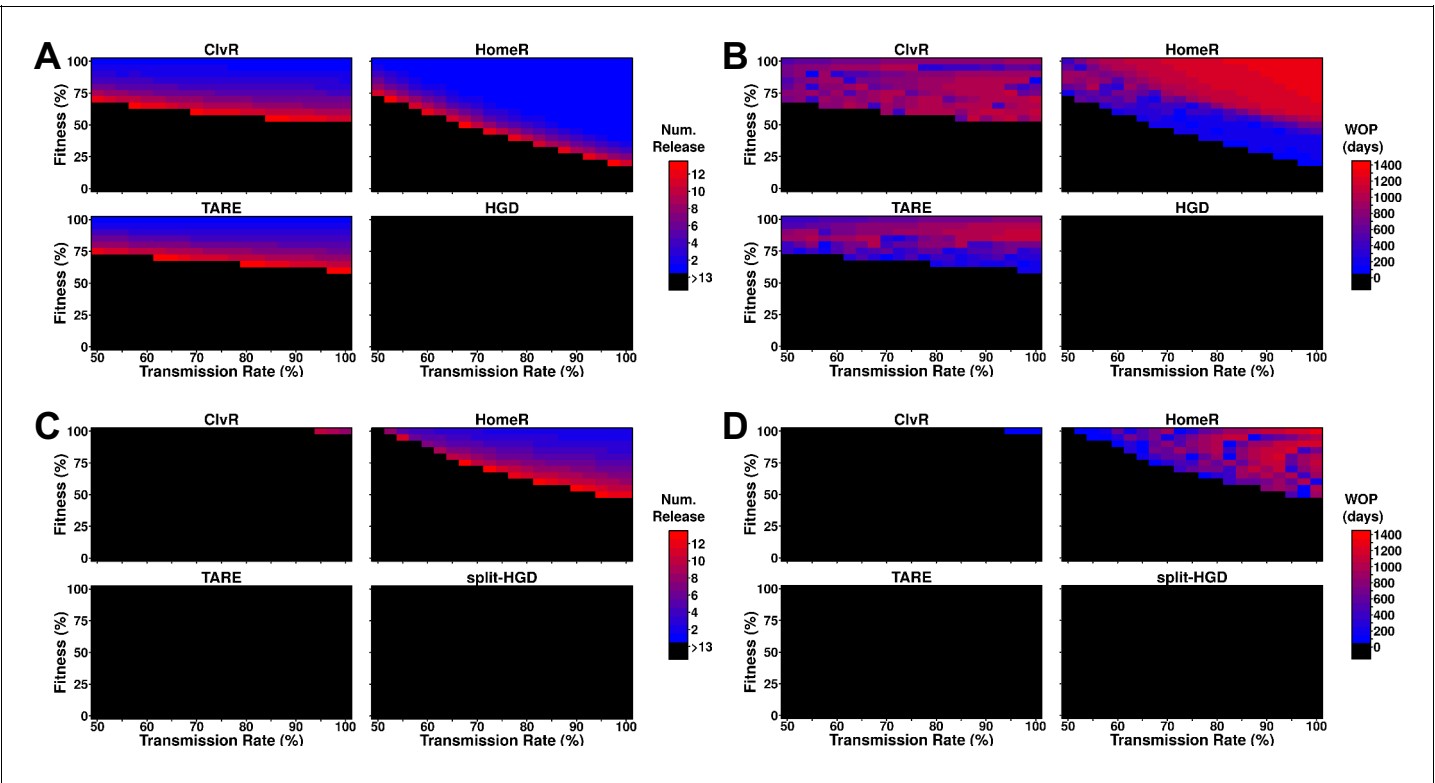

**Figure 4.** Performance of contemporary gene drive systems for population modification with multiple releases. (**A**) Simulations of one-locus designs of ClvR, TARE, HomeR, and HGD in an ecologically consistent model (see 'Materials and methods'). Weekly releases of drive homozygous males (20% of the population size) were performed for up to 13 weeks (3 months, approximately one field season), and the female population was then tested for carrier frequency above 95% at any point within the subsequent 4 years. (**B**) Same setup as (**A**), but now the female populations were measured for how many days the carrier frequency remained above 95%, starting at the first release, lasting up to 4 years. This indicates the window where a disease-refractory allele could provide protection (window-of-protection [WOP]). (**C**) Simulations of two-locus (split-drive) versions of ClvR, TARE, HomeR, and HGD in an ecologically consistent model (see 'Materials and methods'). This time, only the first release was homozygous for Cas9 and the gene drive. Supplementary releases included only Cas9. Male releases were 20% of the total population size, and the female population was measured for drive carrier frequency, not Cas9 frequency, above 95%. (**D**) Using the data from (**C**), we applied the method from (**B**) to measure the WOP, where the drive carrier frequency remained above 95% in the female population. This only measures the drive frequency, not females who carry the Cas9 allele. All simulations contained 100 stochastic repetitions.

## Discussion

We have engineered a system we term HomeR, for population modification that mitigates some issues related to drive resistance. To limit the potential for inducing functional resistant alleles, an essential gene required for insect viability was strategically targeted. Multigenerational population drive experiments indicate that $PolG2^{HomeR}$ can spread and persist efficiently in the presence of Cas9, and this persistence is not impacted by induced resistant alleles, including functional resistant alleles (non-silent R1), mitigating a major challenge for population modification HGDs that are not designed to target essential genes.

The re-coded rescue strategy that we used to develop HomeR was also used in previous *Drosophila* TA non-HGDs (*Champer et al., 2020a*; *Oberhofer et al., 2020a*, *Oberhofer et al., 2020b*, *Oberhofer et al., 2019*; *Figure 3—figure supplement 1A–B*) and recent HGDs in both *Drosophila* (*Champer et al., 2020b*) and *Anopheles stephensi* (*Adolfi et al., 2020*), though each of these examples suffered from potential drawbacks. For example, both the haplolethal HGD (*Champer et al., 2020b*) and the TARE design (*Champer et al., 2020a*) share similar problematic design architectures that can be unstable as they are susceptible to functional resistant alleles induced via recombination between the promoter including sequences 5' of the coding sequence and 3' UTR regions, which are identical between the re-coded sequence and the *wt* sequence (Figure 2—figure supplement 1C in *Champer et al., 2020a* and Figure 2 in *Champer et al., 2020b*). Moreover, a haplolethal HGD (*Champer et al., 2020b*) that biases transmission in females requires a strict germline-specific promoter that limits maternal carryover, otherwise LBM, either mono- or biallelic, may result in dominant negative fitness costs to its carrier (*Figure 3—figure supplement 1E*) and impede drive spread. In fact, our efforts to find such promoters in *Drosophila* proved exceedingly difficult—with previously tested 'germline-specific' promoters such as *nanos* and *vasa* showing significant somatic activity at multiple insertion sites (*Kandul et al., 2020*; *Kandul et al., 2019*). The recent HGD in *A. stephensi* (*Adolfi et al., 2020*) was designed to target and rescue a non-essential gene for viability (i.e. the eye pigmentation *kynurenine hydroxylase* gene), whose disruption was pleiotropic and only partially costly to female fecundity and survival (*Adolfi et al., 2020*; *Gantz et al., 2015*; *Pham et al., 2019*). Notwithstanding, this drive spread efficiently in small, multigenerational laboratory population cages under several release thresholds; however, many drives did not reach, nor maintain, complete fixation presumably due to the viability and partial fertility of drive generated homozygous LOF resistant alleles (*Figure 3—figure supplement 1D*), underscoring the critical importance of targeting a recessive essential gene for such drives especially for larger releases. Comparatively, the ClvR system is quite stable, however, it can be cumbersome to engineer—requiring re-coding of the essential rescue gene, including all target sequences within the coding sequence (lacking introns), and uses an exogenous promoter and 3' UTR, necessitating precise titration of expression from a distal genomic location with exogenous sequences to guarantee rescue without imposing deleterious fitness costs. These features may be difficult to accomplish for essential genes requiring complex regulatory elements and networks not directly adjacent to the target gene. In contrast to the aforementioned drives, (i) HomeR relies on the endogenous promoter sequence of the target gene to facilitate rescue expression which significantly simplifies the design and ensures endogenous expression of the rescue using native regulatory machinery, (ii) creatively designed to target the 3' end of the essential gene to limit the degree of re-coding required for the rescue, (iii) an exogenous 3' UTR to prevent deleterious recombination, and (iv) exploits LBM (*Kandul et al., 2019*) by targeting an essential gene to ensure recessive non-functional resistant alleles result in dominant deleterious/lethal mutations that are actively selected out of a population (*Figure 1—figure supplement 4*), four important design features that distinguish HomeR from other population modification drives.

Our findings are congruent with previous studies demonstrating reduced homing in *Drosophila* males (*Chan et al., 2013*; *Chan et al., 2011*; *Windbichler et al., 2011*). We tested multiple Cas9 lines supporting Cas9 expression in early and/or late germ cells with different levels of specificity and have not achieved high levels of homing as reported in Anopheline mosquito males (*Gantz et al., 2015*; *Kyrou et al., 2018*). Achiasmatic meiosis in *Drosophila* males likely correlates with the weak activity of the HDR pathway (*Preston et al., 2006*), which in turn results in inefficient homing in *Drosophila* males. Mosquito males have chiasmatic meiosis and recombination (*Kitzmiller, 1976*) that require active HDR machinery in primary spermatocytes, possibly contributing to efficient homing. Reduced homing efficacy in *Drosophila* males should be accounted for when

designing HGDs in other species exhibiting achiasmatic meiosis, such as *Drosophila suzukii,* an invasive fruit pest.

Results from independent multigenerational population cage experiments indicate that *HomeR* spreads and persists efficiently in the *nos-Cas9* genetic background (*Figure 2B*). As expected, a single copy of the HomeR inserted at an essential gene provides sufficient rescue and complements the corresponding LOF allele. The *PolG2^HomeR* allele persisted for 10 generations in control cage populations without *Cas9* and its frequency drifted >50%, underscoring the lack of major fitness costs to *PolG2^HomeR*. The LOF alleles complemented by *PolG2^HomeR* also persisted for many generations after a carrier frequency reached 100%. Once LOF alleles are complemented by the *PolG2^HomeR* allele, it takes several generations for LOF alleles to combine as lethal homozygotes and be negatively selected out of the population. The slow-acting elimination of LOF alleles takes especially long time by HGDs targeting non-essential genes or genes whose disruption does not cause complete lethality or sterility of homozygous carriers (*Figure 3—figure supplement 1D*; *Adolfi et al., 2020*; *Gantz et al., 2015*) underscoring the importance of targeting essential genes.

Functional resistant (R1) alleles are a problematic feature shared universally by many kinds of gene drives. These alleles can still be induced even when an essential gene required for insect viability is targeted (*Figure 2—figure supplement 1*). However, it should be noted that here we did not identify any silent R1 mutations (i.e. mutations that change the DNA sequence but not the protein AA sequence) which would be expected to be fitness neutral. Each identified in-frame non-silent *PolG2^R1* allele we found changed at least one AA and thus may still affect the fitness of its carrier, especially since we are targeting an essential gene, preventing such alleles from accumulating at the expense of the drive. Indeed, we observed that functional R1 alleles imposed fitness costs on seven female carriers sampled in drive populations #4 and #5. This fitness cost likely limits their accumulation and results in negative selection out of the population, in favor of the *PolG2^HomeR* alleles, over multiple generations (*Figure 2A–B*) again underscoring the importance of targeting an essential gene. Nevertheless, multiplexing by encoding additional gRNAs into HomeR may further diminish the probability of inducing functional resistant alleles and further increase drive stability, spread, and persistence (*Champer et al., 2018*; *KaramiNejadRanjbar et al., 2018*; *Marshall et al., 2017*; *Oberhofer et al., 2018*).

Splitting HomeR into two genetic loci (*HomeR* and *Cas9*) integrated on different chromosomes serves as an important molecular containment mechanism (*Akbari et al., 2015*; *Long et al., 2020*). The *HomeR* element is able to home into *wt* alleles and bias its transmission. However, the *Cas9* element, which is inherited Mendelianly, is required for its homing. Therefore, the independent assortment of *Cas9* and *HomeR* limits the spread of *HomeR* and acts as a genetic 'brake' for the invasion of HomeR. The spread dynamic of split-HGDs resembles that of high-threshold drives and thus requires a high introduction rate for HomeR to spread into a local population and prevents its spread into neighboring populations, which is an important feature for confining drive spread and may be necessary for initial field testing of gene drives (*Adelman et al., 2017*; *Akbari et al., 2015*; *Friedman et al., 2020*; *Kandul et al., 2020*; *Li et al., 2020*; *Raban and Akbari, 2017*; *Raban et al., 2020*). Moreover, HomeR can be further confined by fitness costs to either the HomeR drive itself or to the *Cas9* element, and our experiments revealed that the Cas9 element imposed significant fitness costs that can impede drive invasion (*Figure 2C*). Notwithstanding, even with significant fitness costs, multiple releases of the HomeR could still enable drive spread and long-term persistence as evidenced by mathematical models (*Figure 4*). As an additional safety measure, if unintended consequences arise, HomeR's spread can be reversed by reintroduction of insects harboring *wt* alleles of the gene targeted. Notwithstanding, if desired, HomeR could facilely be converted into a non-localized gene drive by incorporating the Cas9 into the HomeR drive cassette and our modeling illustrates that it could perform quite well under this configuration (*Figure 3A,B*, *Figure 4A,B*). Taken together, the split-drive design of HomeR is a safe localized gene drive technology that could be widely adopted and implemented for local population control, and if a non-localized drive is desired for more wide scale spread, HomeR could be adapted for that purpose too.

In sum, HomeR combines promising aspects of current population modification drives—confinability, high transmission of HGDs, and resilience to EJ generated resistant alleles (R2 type and R1 type that induces a fitness cost) similar to TA drives (*Figure 3—figure supplement 1*). Modeling illustrates success of both design aspects in linked or split-drive form, demonstrating robust behavior over a range of parameter combinations (*Figures 3–4*). This underscores its stability and resilience

to EJ alleles, overcoming a significant hurdle for current HGD designs. Given the simplicity of the HomeR design, it could be universally adapted to a wide range of species including human disease vectors in the future.

# Materials and methods

## Key resources table

| Reagent type (species) or resource | Designation | Source or reference | Identifiers | Additional information |
|---|---|---|---|---|
| Strain, strain background (*D. melanogaster*) | gRNA#1$^{PolG2}$ | 159674 | 91378 | This publication |
| Strain, strain background (*D. melanogaster*) | gRNA#2$^{PolG2}$ | 159675 | n/a | This publication |
| Strain, strain background (*D. melanogaster*) | HomeR$^{PolG2}$ | 159676 | Gene drives cannot be deposited at BDSC | This publication |
| Strain, strain background (*D. melanogaster*) | HomeR(B)$^{PolG2}$ | 159677 | Gene drives cannot be deposited at BDSC | This publication |
| Strain, strain background (*D. melanogaster*) | nos-Cas9 | 112685 | 79004 | 30622266 |
| Strain, strain background (*D. melanogaster*) | vas-Cas9 | 112686 | 79005 | 30622266 |
| Strain, strain background (*D. melanogaster*) | Uniq-Cas9 | 112687 | 79006 | 30622266 |
| Strain, strain background (*D. melanogaster*) | Act5C-Cas9 | n/a | 54590 | 25002478 |
| Strain, strain background (*D. melanogaster*) | exuL-Cas9 | 159671 | 91375 | This publication |
| Strain, strain background (*D. melanogaster*) | Rcd1r-Cas9 | 159673 | 91377 | This publication |
| Strain, strain background (*D. melanogaster*) | bTub-Cas9 | 159672 | 91376 | This publication |

## Selection of Cas9/gRNA target sites

We inserted a HomeR in *DNA Polymerase gamma subunit 2* (*PolG2* or *Pol-γ35*, CG33650). *PolG2* is an essential gene required for insect viability. The C-terminal domain of *PolG2* is located at the end of the coding sequence, which facilitates its re-coding (*Figure 1—figure supplement 1A–B*). We PCR-amplified a 413-base fragment of the domain with 1073A.S1F and 1073A.S2R from multiple *Drosophila* strains (*w$^{1118}$*, Canton S, Oregon R, *nos-Cas9*; *Kandul et al., 2019*) and used the consensus sequence along with the tool CHOPCHOP v2 (*Labun et al., 2016*) to choose two gRNA targets sites that minimize off-target cleavage. In addition, we used the DGRP2 (http://dgrp2.gnets.ncsu.edu) that includes natural variation in genome architecture among 205 *D. melanogaster* genetic reference panel lines (*Huang et al., 2014*; *Mackay et al., 2012*) to explore SNPs found inside both gRNA target sequences.

## Design and assembly of genetic constructs

We used Gibson enzymatic assembly to build all genetic constructs (*Gibson et al., 2009*). To assemble both gRNA constructs, we used the previously described *sgRNA^Sxl* plasmid (*Kandul et al., 2019*; Addgene #112688) harboring the mini-*white* gene and attB docking site. We removed the fragment encompassing the U6.3 promoter and gRNA scaffold by AscI and SacII digestion, and cloned it back as two fragments overlapping at a novel gRNA sequence (*Figure 1—figure supplement 1A*). Both *gRNA#1^PolG2* and *gRNA#2^PolG2* plasmids targeting *PolG2* are deposited at http://www.addgene.org/ (#159774 and #159675).

We assembled two *HomeR^PolG2* constructs using two tested gRNAs (*Figure 1—figure supplement 2A*). Each *HomeR^PolG2* was built around a specific gRNA, with matching LHA and RHA: *HomeR^PolG2* harbored *gRNA#1^PolG2*, and *HomeR(B)* had *gRNA#2^PolG2*. We digested the *nos-Cas9* plasmid (*Kandul et al., 2019*; Addgene #112685) with AvrII and AscI, preserving the backbone containing the *piggyBac* left and right sequences that encompass the *Opie-dsRed-SV40* marker gene. The HomeR construct was assembled between *Opie-dsRed-SV40* and *piggyBacR* in three steps. First, we cloned the *gRNA#1* or *#2* from the corresponding plasmid together with the *3xP3-eGFP-SV40* marker gene, to tag site-specific insertion of *GDe*. Then, we cloned three fragments: (i) LHA, which was amplified from the *Drosophila* genomic DNA; (ii) the re-coded fragment downstream from the gRNA cut site, which was PCR-amplified from the dePolG2 gBlock custom synthesized by IDT (*Supplementary file 1*); (iii) the p10 3' UTR to provide robust expression (*Pfeiffer et al., 2012*) of the re-coded *PolG2* rescue. Finally, we cloned RHA, which was PCR-amplified from genomic DNA, corresponding to each specific gRNA cut site. Importantly, the re-coding was carefully designed to ensure the translation of the re-coded DNA sequence in the *wt* amino acid sequence of *Pol2* with respect to *Drosophila* codon usage bias. Both *HomeR^PolG2* and *HomeR(B)^PolG2* plasmids, targeting the *PolG2* locus, are deposited at http://www.addgene.org/ (#159676 and #159677).

To assemble the three constructs for testis-specific Cas9 expression, we used a plasmid harboring the *hSpCas9-T2A-GFP*, the *Opie2-dsRed* transformation marker, and both *piggyBac* and attB-docking sites, which were previously used to establish Cas9 transgenic lines in *Aedes aegypti* (*Li et al., 2017*) and *D. melanogaster* (*Kandul et al., 2020*; *Kandul et al., 2019*). We removed the *Ubiquitin 63E* promoter from the *ubiq-Cas9* plasmid (Addgene #112686) (*Kandul et al., 2019*) by digesting it with SwaI at +27°C and then with NotI at +37°C, and cloned a promoter fragment amplified from the *Drosophila* genomic DNA. The *Drosophila exuperantia* (CG8994) 783 bp fragment (*exuL*) upstream of the *exuperantia* gene was amplified with ExuL.1F and ExuL.2R primers (*Supplementary file 1*) and cloned to assemble the *exuL-Cas9* plasmid. The *Rcd-1 related* (*Rcd1r*, CG9573; *Chan et al., 2013*) and *β-Tubulin 85D* (*βTub*; *Chan et al., 2011*; *Michiels et al., 1989*) promoters support early and late, respectively, testis-specific expression in *Drosophila* males. The 937-base-long fragment upstream of *Rcd1r* was amplified with 1095.C1F and 1095.C2R primers and cloned to assemble the *Rcd1r-Cas9* plasmid. The 481-base-long fragment upstream of *βTub* was amplified with βTub.1F and βTub.2R primers (*Supplementary file 1*) and cloned to build the *βTub-Cas9* plasmid. Three plasmids for testis-specific Cas9 expression are deposited at http://www.addgene.org/ (#159671–159773).

## Fly maintenance and transgenesis

Flies were maintained under standard conditions: 26°C with a 12 hr/12 hr light/dark cycle. Embryo injections were performed by Rainbow Transgenic Flies, Inc. We used φC31-mediated integration (*Groth et al., 2004*) to insert the *gRNA#1* and *gRNA#2* constructs at the P{CaryP}attP1 site on the second chromosome (BDSC #8621), and the *exuL-Cas9*, *βTub-Cas9*, and *Rcd1r-Cas9* constructs at the PBac{y+-attP-3B}KV00033 on the third chromosome (BDSC #9750). Two methods were used to generate the site-specific insertion of *HomeR^PolG2* or *HomeR(B)^PolG2* constructs at the *gRNA#1^PolG2* or *gRNA#2^PolG2* cut sites, respectively, inside the *PolG2* gene via HDR. First, we injected the mixture of HomeR and helper *phsp-pBac*, carrying the piggyBac transposase (*Handler and Harrell, 1999*), plasmids (500 and 250 ng/μl, respectively, in 30 μl) into *w^1118* embryos. Random insertions of *HomeR^PolG2* and *HomeR(B)^PolG2*, assessed by double (eye-specific GFP and body-specific dsRed) fluorescence (*Figure 1—figure supplement 2B*), established with this injection were genetically crossed to *nos-Cas9/nos-Cas9* (BDSC #79004; *Kandul et al., 2019*) flies to 'relocate' *HomeR^PolG2* or *HomeR(B)^PolG2* to the corresponding gRNA cut site via HACK (*Lin and Potter, 2016*). A few site-specific *PolG2^HomeR* and *PolG2^HomeR(B)* lines tagged with only eye-specific GFP fluorescence were recovered.

Second, we injected *HomeR^PolG2^* or *HomeR(B)^PolG2^* plasmids directly into *nos-Cas9/nos-Cas9* (BDSC #79004; *Kandul et al., 2019*) embryos, generating multiple independent, site-specific insertions for each *PolG2^HomeR^* (*Figure 1—figure supplement 2B*). Recovered transgenic lines were balanced on the second and third chromosomes using single-chromosome balancer lines (*w^1118^; CyO/sna^Sco^* for II and *w^1118^; TM3, Sb^1^/TM6B, Tb^1^* for III) or a double-chromosome balancer line (*w^1118^; CyO/Sp; Dr/ TM6C, Sb^1^*). While both techniques (random insertion/HACK and HDR) worked to generate site-directed insertions, all subsequent analysis was performed exclusively on lines derived from the HDR-based transgenesis approach.

We established three homozygous lines of *PolG2^HomeR^* and *PolG2^HomeR(B)^* from independent insertion lines, and confirmed the precision of site-specific insertions by sequencing the borders between HomeR constructs and the *Drosophila* genome (*Figure 1—figure supplement 2C*). The 1118-base-long fragment overlapping the left border was PCR-amplified with 1076B.S9F and 1076B. S2R and was sequenced with 1076B.S3F and 1076B.S4R primers. The same-length fragment at the right border was amplified with 1073A.S1F and 1076B.S10R and was sequenced with 1076B.S7F and 1076B.S8R primers (*Supplementary file 1*).

## Fly genetics and imaging

Flies were examined, scored, and imaged on a Leica M165FC fluorescent stereomicroscope equipped with a Leica DMC2900 camera. We assessed the transmission rate of HomeR by following its eye-specific GFP fluorescence, while the inheritance of *Cas9* was tracked via body-specific dsRed fluorescence (*Figure 2A*). All genetic crosses were done in fly vials using groups of 10 males and 10 females.

## RNA^PolG2^ cleavage assay

To assess the cleavage efficiency of each gRNA targeting the C-terminal domain of *PolG2*, we genetically crossed 10 *w^1118^; gRNA#1^PolG2^* or *w^1118^; gRNA#2^PolG2^* homozygous males to 10 *y^1^, Act5C-Cas9, w^1118^, Lig4* (*Zhang et al., 2014*) (BDSC #58492) homozygous females, and scored the lethality of F$_1$ males (*Figure 1—figure supplement 1C*). The F$_1$ males would then inherit the X chromosome from their mothers, expressing *gRNA#1^PolG2^* or *gRNA#2^PolG2^* with *Act5C-Cas9* in a *Lig4*-null genetic background, and this results in male lethality when a tested gRNA directs cleavage of the *PolG2* locus. To assess the induced lethality in the *Lig4+/+* genetic background, we crossed 10 *y^1^, Act5C-Cas9, w^1118^* (BDSC #54590; *Port et al., 2014*) flies to 10 *U6.3-gRNA#1^PolG2^* flies in both directions, and scored survival of trans-heterozygous and heterozygous F$_1$ progeny. To measure the Cas9/gRNA-directed cleavage of *PolG2* by maternally deposited Cas9 protein in the *Lig4+* background, the same homozygous males were genetically crossed to *w^1118^/w^1118^; nos-Cas9/CyO* females (*Figure 1—figure supplement 1D*), and the F$_1$ progeny, harboring *gRNA#1^PolG2^* or *gRNA#2^PolG2^*, were scored and compared to each other.

## Assessment of *PolG2^HomeR^* transmission rates

To compare transmission rates of *PolG2^HomeR^* and *PolG2^HomeR(B)^*, we first established trans-heterozygous parent flies by genetically crossing *PolG2^HomeR^/PolG2^HomeR^; +/+* or *PolG2^HomeR(B)^/PolG2^HomeR(B)^; +/+* females to *+/+; nos-Cas9/nos-Cas9* males. We then assessed the transmission rates by trans-heterozygous parent females and males crossed to *wt* flies. For controls, we estimated the transmission rates of *HomeR^PolG2^* and *HomeR(B)^PolG2^* in the absence of Cas9, by heterozygous *PolG2^HomeR^/+* or *PolG2^HomeR(B)^/+* females and males crossed to *wt* flies (*Figure 1—figure supplement 3*). To explore the effect of maternally deposited Cas9 protein on transmission of *PolG2^HomeR^*, we generated heterozygous *PolG2^HomeR^/CyO* embryos containing Cas9 protein deposited by *nos-Cas9/CyO* mothers and estimated the transmission of *PolG2^HomeR^* by females and males raised from these embryos and crossed to *wt* flies. We tested five different Cas9 lines—supporting germline (*vas-Cas9*), ubiquitous (*ubiq-Cas9, Act5C-Cas9*), and early (*exuL-cas9, Rcd1r-Cas9*) or late testes-specific expression (*βTub-Cas9*)—together with the strongest HomeR, *PolG2^HomeR^*. To control for position effect variegation, each *Cas9* transgene was inserted at the same attP docking site on the third chromosome, except for *Act5C-Cas9* that was integrated on the X chromosome (*Port et al., 2014*). Ten trans-heterozygous females or males, generated by crossing homozygous *PolG2^HomeR^* females to

homozygous *Cas9* males, were genetically crossed to *wt* flies and the transmission of *PolG2^HomeR^* was quantified in their $F_1$ progeny (*Figure 1D*).

## Egg hatching and egg-to-adult survival rates

To identify the mechanism of the super-Mendelian transmission of *PolG2^HomeR^*, we assessed the percentage of $F_1$ hatched eggs laid by trans-heterozygous *PolG2^HomeR^/+; nos-Cas9/+* females genetically crossed to *wt* males and compared it to those hatched from two types of heterozygous females: *PolG2^HomeR^/+; +/+* ♀ and *+/+; nos-Cas9/+* ♀ (*Figure 1B*). We collected virgin females and aged them for 3 days inside food vials supplemented with a yeast paste, then five groups of 25 virgin females of each type were transferred into vials with fresh food containing 25 *wt* males and allowed to mate overnight (12 hr) in the dark. Then, all males were removed from the vials, while females were transferred into small embryo collection cages (Genesee Scientific 59–100) with grape juice agar plates. After 12 hr of egg laying, a batch of at least 200 laid eggs was counted for each sample group and incubated for 24 hr at 26°C before the number of unhatched eggs was counted. To assess the egg-to-adult survival rate, at least 12 groups of 75 eggs were collected to each type of tested progeny and transferred to individual vials. The emerged flies from each vial were counted (*Figure 1C*), and their sex and fluorescence were scored (*Supplementary file 7*).

## 'Fishing' for functional resistant alleles, *PolG2^R1^*

To explore the generation and accumulation of functional resistant alleles induced by EJ, we initiated three populations by crossing 50 *+/+; nos-Cas9/nos-Cas9* females and 50 *PolG2^HomeR^/PolG2^HomeR^; nos-Cas9/nos-Cas9* (*Figure 2A*) males in 0.3 l plastic bottles (VWR *Drosophila* Bottle 75813–110). Parent (P) flies were removed after 6 days, and their progeny were allowed to develop, eclose, and mate for 13–15 days. This established a 100% heterozygous *PolG2^HomeR^/+; nos-Cas9/+* population in every bottle at the next generation ($G_0$). Each generation, around 250–350 emerged flies were anesthetized using $CO_2$, and their genotypes with respect to *PolG2^HomeR^* (presence or absence) were determined using the dominant eye-specific GFP marker. Then they were transferred to a fresh bottle and allowed to lay eggs for 6 days before removing them, and the cycle was repeated. Three populations were maintained in this way for 11 generations, which corresponds to 10 generations of gene drive. Note that any fly scored without the *PolG2^HomeR^* allele was transferred into a fresh bottle to ensure any *PolG2* resistant or *wt* alleles could be passed to the next generation. We retrieved and froze the flies for genotyping only after 6days to ensure sufficient time for breeding. We expected that if *PolG2^R1^* alleles were frequently generated and did not incur fitness costs, they would persist and accumulate over a few generations at the expense of *PolG2^HomeR^*. However, as we did not find any fly without the *PolG2^HomeR^* allele after $G_3$, we stopped populations after 10 generations and froze 60 flies after $G_{10}$ for further sequence analysis.

## HomeR population drives in the *Cas9* and *wt* genetic backgrounds

For HomeR drives in the *nos-Cas9/nos-Cas9* genetic background, we seeded five experimental (*Cas9+*) drives and three control (*Cas9–*) drives with 50 homozygous *PolG2^HomeR^/PolG2^HomeR^* males and 50 *wt* males together with 100 *wt* virgin females in 0.3 l plastic bottles. Seeded flies either encoded Cas9 (experimental drive) or not (control or 'no-drive', *Figure 2B*). For HomeR drives in the *wt* genetic background, we seeded four drive populations with 100 *wt* virgin females and double homozygous (*PolG2^HomeR^/PolG2^HomeR^; nos-Cas9/nos-Cas9*) males mixed with *wt* males at the ratios of 1:1 (two populations with 25% of *PolG2^HomeR^*) or 3:1 (two populations with 37.5% of *PolG2^HomeR^*, *Figure 2C*). Note that the *PolG2^HomeR^* males were competing with *wt* males for female mates, and their mating competitiveness could be scored by the dominant 3xP3-GFP marker of *PolG2^HomeR^* in their progeny at generation 0 ($G_0$). Both types of homozygous *PolG2^HomeR^* males with and without *Cas9* were able to compete with the corresponding *wt* males for female mates resulting in the increase of *PolG2^HomeR^* from 25% or 37.5% in parents to nearly 50% or 70%, respectively, at generation 0 (*Figure 2B–C*). The discrete-generation populations were maintained and scored as described above. Each generation, around 250–350 emerged flies were anesthetized using $CO_2$, and their genotypes were scored for the presence or absence of *PolG2^HomeR^* (eye-specific GFP) and *nos-Cas9* (body-specific dsRed). Then they were transferred to a fresh bottle and allowed to lay eggs for 6 days before removing them, and the cycle was repeated.

## Sequencing of induced resistant alleles

To analyze the molecular changes that caused functional in-frame (R1) and LOF (R2) resistant mutations in *PolG2*, we PCR-amplified the 232-base-long genomic region containing both *gRNA#1$^{PolG2}$* and *gRNA#2$^{PolG2}$* cut sites using 1073A.S3F and 1073A.S4R primers (*Supplementary file 1*). For PCR genotyping from a single fly, we followed the single-fly genomic DNA prep protocol (*Kandul et al., 2019*). PCR amplicons were purified using the QIAquick PCR purification kit (QIA-GEN), subcloned into the pCR2.1-TOPO plasmid (Thermo Fisher), and at least seven clones were sequenced in both directions by Sanger sequencing at Retrogen and/or Genewiz to identify both alleles in each fly. Sequence AB1 files were aligned against the corresponding *wt* sequence of *PolG2* in SnapGene 4.

To explore the diversity of resistant alleles persisting after 10 generations of *PolG2$^{HomeR}$* in a 100% heterozygous population, we froze 60 flies (30 ♀ and 30 ♂), each harboring at least one copy of the dominant marker of *PolG2$^{HomeR}$*, from each lineages after $G_{10}$ (*Figure 2A*). Using these flies, we quantified any resistant and *wt* alleles remaining in the population via Illumina sequencing of heterogeneous PCR amplicons at the *PolG2* locus. Note that PCR amplicons did not include the *PolG2$^{HomeR}$* allele due to its length (*Figure 1A*). Additionally, this assay will not be able to accurately distinguish between germline and somatic mutations as whole flies were used. DNA was extracted using the DNeasy Blood and Tissue Kit (QIAGEN). To analyze heterogeneous PCR products, we used the Amplicon-EZ service by Genewiz and followed the Genewiz guidelines for sample preparation. In brief, Illumina adapters were added to the 1073A.S3F and 1073A.S4R primers to simplify the library preparation, PCR products were purified using QIAquick PCR purification kit (QIAGEN), around 50,000 one-direction reads covering the entire amplicon length were generated, and relative abundances of recovered SBS and *indel* alleles at the *gRNA#2$^{PolG2}$* cut site were inferred using Galaxy tools (*Afgan et al., 2018*). Amplicon-EZ data from Genewiz were first uploaded to Galaxy.org. A quality control was performed using FASTQC. Sequence data were then paired and aligned against the *PolG2$^{WT}$* sequence using Map with BWA-MEM under 'Simple Illumina mode'. The SBS and *indel* alleles were detected using FreeBayes, with the parameter selection level set to 'simple diploid calling'.

## Model fitting to cage experiment data

Empirical data from the *HomeR* population replacement experiments were used to parameterize a model of CRISPR-based homing gene drive including resistant allele formation. Model fitting was carried out for all five gene drive cage experiments using Markov chain Monte Carlo (MCMC) methods in which estimated parameters related to cleavage efficiencies in females and males, accurate HDR frequencies given cleavage in females and males, the proportion of resistant alleles that are in-frame and cost-free, and the fitness cost associated with having the *HomeR* system. We considered discrete generations, random mixing, and Mendelian inheritance rules at the gene drive locus, with the exception that for adults heterozygous for the homing allele (denoted by 'H') and *wt* allele (denoted by 'W'), a proportion, *c*, of the W alleles are cleaved, while a proportion, 1 *c*, remain as W alleles. Of those that are cleaved, a proportion, $p_{HDR}$, are subject to accurate HDR and become H alleles, while a proportion, (1-$p_{HDR}$), become resistant alleles. Of those that become resistant alleles, a proportion, $p_{RES}$, become in-frame, functional, cost-free resistant alleles (denoted by 'R'), while the remainder, (1-$p_{RES}$), become out-of-frame, non-functional, or otherwise costly resistant alleles (denoted by 'B'). The values of *c* and $p_{HDR}$ were allowed to vary depending on whether the HW individual is female or male. The fitness cost associated with the *HomeR* system, $s_{H,F}$, was assumed to be female-specific. These considerations allowed us to calculate expected genotype frequencies in the next generation, and to explore the parameter values that maximize the likelihood of the experimental data. The model fitting framework is described in full in S1 text of *Pham et al., 2019*.

## Comparative modeling of gene drive systems

Comparative gene drive simulations were performed using a discrete-generation version of the Mosquito Gene Drive Explorer (MGDrivE) modeling framework (*Sanchez et al., 2019*). The first generation was seeded with 400 adults, 50% *wt* females, 25% *wt* males, and 25% homozygous gene drive males. At each generation, adult females mate with males, thereby obtaining a composite mated genotype (their own, and that of their mate) with mate choice following a multinomial distribution

determined by adult male genotype frequencies. Egg production by mated adult females then follows a Poisson distribution, proportional to the genotype-specific lifetime fecundity of the adult female. Offspring genotype follows a multinomial distribution informed by the composite mated female genotype and the inheritance pattern of the gene drive system. Sex distribution of offspring follows a binomial distribution, assuming equal probability for each sex. Female and male adults from each generation are then sampled equally to seed the next generation, with a sample size of 400 individuals (200 female and 200 male), following a multivariate hypergeometric distribution. Twenty-five repetitions were run for each drive in the trace plots (*Figure 3A and C*), and 100 repetitions were run for each parameter combination in the heatmaps (*Figure 3B and D*).

The inheritance pattern is captured by the 'inheritance cube' module of MGDrivE (*Sanchez et al., 2019*). ClvR and TARE constructs were implemented to match their published descriptions (*Champer et al., 2020a*; *Oberhofer et al., 2020a*, *Oberhofer et al., 2019*). HomeR and HGD were implemented as one- or two-locus systems following equivalent inheritance rules. When Cas9 and gRNAs co-occur in the same individual, *wt* alleles are cleaved at a rate $c_F$ ($c_M$) (female- (male-) specific cleavage), with 1-$c_F$ (1 $c_M$) remaining *wt*. Given cleavage, successful HDR occurs at a rate $ch_F$ ($ch_M$), with 1-$ch_F$ (1-$ch_M$) alleles undergoing some form of EJ. Of these, a proportion, $cr_F$ ($cr_M$), are in-frame EJ alleles, while the remainder, 1-$cr_F$ (1-$cr_M$), are LOF alleles. Maternal carryover (maternal deposition, or maternal perdurance) was modeled to occur in zygotes of mothers having both Cas9 and gRNAs, impacting a proportion, $d_F$, of zygotes. Of the *wt* alleles in impacted zygotes, a proportion, $dr_F$, become in-frame EJ alleles, while the remainder, 1-$dr_F$, become LOF alleles. These inheritance rules apply to both HomeR and HGD, with differing fitness costs.

ClvR (*Oberhofer et al., 2020a*; *Oberhofer et al., 2019*) was modeled using a 99% cleavage rate in female and male germ cells, as well as in embryos from maternal carryover. For two-locus ClvR, the two loci were assumed to undergo independent assortment ($\geq$50 cM separation), as was assumed for all two-locus systems in this analysis. For both configurations, it was assumed that 0.1% of cleaved alleles were converted to functional resistant alleles (R1 type), and the rest became LOF alleles (R2 type). In addition to the 50% egg-hatching reduction due to the non-homing drive (*Figure 3—figure supplement 1A–B*), an additional 5% reduction in fecundity was applied to females that harbored Cas9. For consistency, TARE, HGD, and HomeR (for ideal parameters) also used a cleavage rate of 99% in females and males, though TARE demonstrated lower maternal carryover (*Champer et al., 2020a*), and was modeled with 95% cleavage. HGD and HomeR (for ideal parameters), which rely on HDR, were simulated with 90% HDR rates in females and males. Cleaved alleles that did not undergo HDR were assumed to be R1 alleles with proportion 0.5%, and R2 LOF alleles the remainder of the time. TARE and HomeR were also modeled with a small (5%) fitness reduction, applied as a reduction of female fecundity. Since an HGD does not provide a rescue for a disrupted target gene, its carriers demonstrate higher fitness costs and were assigned a 20% fitness reduction with the assumption that the HGD is inserting into a non-lethal gene that imposes a low/moderate fitness cost. Experimentally derived parameters for HomeR differed from ideal parameters in two ways: (i) there was no HDR in males (although cleavage remained the same) and (ii) 1% of EJ-repaired *wt* alleles were converted into R1 alleles (cf. 0.5% for the ideal case).

To determine the number of releases required to introgress effector genes into 95% of the female portion of a population, and ascertain how long that introgression could be effective (up to 4 years; *Figure 4*), we performed simulations using the full version of MGDrivE (*Sanchez et al., 2019*), implementing overlapping generations and density-dependent growth effects on aquatic stages. All gene drive characteristics were maintained as stated above. For one-locus designs, male releases, up to 13, were performed at 20% of the total population size (10,000). For two-locus designs, the first release was males homozygous for Cas9 and gene drive, but subsequent releases were only homozygous for Cas9. The two-locus releases were 20% of the total population size. Life cycle parameters are: 2 days for egg maturation, 5 days for larval maturation, 1 day for pupal maturation, and an expected adult lifespan of 11 days (*Li et al., 2020*). All simulations were performed, analyzed, and plotted in R (*R Development Core Team, 2017*). Code is available upon request.

## Statistical analysis

Statistical analysis was performed in JMP 8.0.2 by SAS Institute Inc, and graphs were constructed in Prism 8.4.1 for MacOS by GraphPad Software LLC. At least three biological replicates were used to

generate statistical means for comparison. p-values were calculated using a two-sample Student's *t* test with equal or unequal variance.

## Gene drive safety measures

All gene drive crosses were performed in accordance with protocols approved by the Institutional Biosafety Committee at UCSD, in which gene drive experiments were performed in a high-security ACL2 barrier facility in plastic vials that were autoclaved prior to being discarded, in accordance with currently suggested guidelines for the laboratory confinement of gene drive systems (*Akbari et al., 2015*; *National Academies of Sciences, Engineering, and Medicine et al., 2016*).

## Ethical conduct of research

We have complied with all relevant ethical regulations for animal testing and research and conformed to the UCSD institutionally approved biological use authorization protocol (BUA #R2401).

## Acknowledgements

This work was supported in part by funding from a DARPA Safe Genes Program Grant (HR0011-17-2-0047), and NIH awards (R21RAI149161A, R01AI151004, DP2AI152071) awarded to OSA. The functional characterization of the *ExuL* promoter was done by OSA while at Caltech working with Bruce A Hay.

## Additional information

### Competing interests

Nikolay P Kandul: is a consultant for Agragene. Omar S Akbari: is a founder of Agragene, Inc, has an equity interest, and serves on the company's Scientific Advisory Board. The other authors declare that no competing interests exist.

### Funding

| Funder | Grant reference number | Author |
|---|---|---|
| Defense Advanced Research Projects Agency | HR0011-17-2-0047 | Omar S Akbari |
| National Institutes of Health | R21RAI149161A | Omar S Akbari |
| National Institutes of Health | R01AI151004 | Omar S Akbari |
| National Institutes of Health | DP2AI152071 | Omar S Akbari |

The funders had no role in study design, data collection and interpretation, or the decision to submit the work for publication.

### Author contributions

Nikolay P Kandul, Conceptualization, Formal analysis, Supervision, Investigation, Methodology, Writing - original draft, Writing - review and editing; Junru Liu, Data curation, Investigation, Writing - review and editing; Jared B Bennett, Formal analysis, Investigation, Visualization, Methodology, Writing - review and editing; John M Marshall, Formal analysis, Supervision, Funding acquisition, Investigation, Methodology, Writing - review and editing; Omar S Akbari, Conceptualization, Formal analysis, Supervision, Funding acquisition, Investigation, Methodology, Writing - original draft, Project administration, Writing - review and editing

### Author ORCIDs

Nikolay P Kandul (iD) https://orcid.org/0000-0001-7347-5558
Jared B Bennett (iD) http://orcid.org/0000-0003-4718-257X
John M Marshall (iD) http://orcid.org/0000-0003-0603-7341
Omar S Akbari (iD) https://orcid.org/0000-0002-6853-9884

Decision letter and Author response
Decision letter https://doi.org/10.7554/eLife.65939.sa1
Author response https://doi.org/10.7554/eLife.65939.sa2

## Additional files

### Supplementary files

• Supplementary file 1. Target sequence of gRNA and primers used in this study.

• Supplementary file 2. Cleavage assay of two gRNAs$^{PolG2}$ with Act5C-Cas9 in the Lig4Δ genetic background.

• Supplementary file 3. Cleavage assay of two gRNAs$^{PolG2}$ with nos-Cas9 and Act5C-Cas9.

• Supplementary file 4. Transmission rate of PolG2$^{HomeR}$ in conjunction with different Cas9 lines.

• Supplementary file 5. Transmission rate of PolG2$^{HomeR(B)}$ in conjunction with nos-Cas9.

• Supplementary file 6. Hatching rate of eggs laid by PolG2$^{HomeR}$ /+; nos-Cas9/+ females.

• Supplementary file 7. Egg-to-adult survival of PolG2$^{HomeR}$ /+; nos-Cas9/+ females' progeny.

• Supplementary file 8. Induced resistant alleles in PolG2$^{HomeR}$ /+; nos-Cas9/nos-Cas9 flies over 10 generations.

• Supplementary file 9. Experimental drives of PolG2$^{HomeR}$ in the nos-Cas9/nos-Cas9 homozygous genetic background.

• Supplementary file 10. Control drives of PolG2$^{HomeR}$ without the Cas9 transgene.

• Supplementary file 11. Experimental drives of PolG2$^{HomeR}$ in conjunction with nos-Cas9 into the wt genetic background.

• Transparent reporting form

### Data availability

All data are represented fully within the tables and figures. The gRNA#1PolG2, gRNA#2PolG2, HomeRPolG2, HomeR(B)PolG2, exuL-Cas9, Rcd1r-Cas9, and βTub-Cas9 plasmids and corresponding fly lines are deposited at http://www.addgene.org/ (159671-159677) and the Bloomington *Drosophila* Stock Center (91375-91378), respectively.

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
