## [Decision Letter]

**Acceptance summary:**

The paper describes and tests a gene drive system by inserting a synthetic rescue gene into an essential gene that also contains the gRNA (gene drive) but not the Cas9 (safe split gene drive). Although there are clearly some limitations to the effectiveness of the drive, to a large extent due to the fitness cost of Cas-9, this new strategy will be a useful path to follow in order to thwart evolution of resistance to the drive. This will move the field forward in getting researchers to broaden their efforts in development of gene drives for specific purposes.

**Decision letter after peer review:**

[Editors’ note: the authors submitted for reconsideration following the decision after peer review. What follows is the decision letter after the first round of review.]

Thank you for submitting your work entitled "A home and rescue gene drive efficiently spreads and persists in populations" for consideration by *eLife*. Your article has been reviewed by a Senior Editor, a Reviewing Editor, and four reviewers. The following individual involved in review of your submission has agreed to reveal their identity: Ernst A Wimmer (Reviewer #2).

Our decision has been reached after consultation between the reviewers. Based on these discussions and the individual reviews below, we feel that the paper is of significant conceptual and practical interest but it is lacking in several points that should absolutely be addressed before the paper can be considered. I therefore we regret to inform you that your work will not be considered for publication in *eLife* but we strongly encourage you to submit a new manuscript once these points have been addressed carefully, which should take a significant amount of time. I wish to emphasize that the reviewers would like this paper to be published in *eLife* but it needs this additional work. The reviewers all also strongly believe that the presentation of the paper needs to be seriously fixed to make it simpler and targeted to a wide audience, which will be very interested by these results.

The following points must be addressed as they are the two main points the manuscript is trying to make but they are not well supported:

– The major claim of the paper, the release of the drive into a wild type population with no Cas9 must be investigated as there is no practical demonstration of a split drive.

– There is also no final demonstration of the repression of R1 alleles: The strategy for sampling resistance appears to not be appropriate and must be changed as suggested by reviewer #4.

– The paper claims to have used an ultra conserved gene. As so much importance is placed on the drive design overcoming resistance, this must be demonstrated. A gene that is considered as ultra conserved would require that there is little or no nucleotide variation (e.g. *dsx* in the Kyrou et al.,). Conservation at the protein level does little to protect against synonymous mutants that would constitute resistance. A better justification for this choice and a demonstration that it was indeed the right choice must be presented.

– The data are really presented in a very difficult way and the paper must be extensively revised with a much broader audience in mind.

– The reviewers did a very thorough review of the paper that should help you to improve it. The individual reviews are included below.

Reviewer #1:

This paper shows that Cas 9 mediated homology directed repair can be used to insert at synthetic rescue gene into an essential gene, here mitochondrial *Pol-γ35* was chosen. The insertion is marked by an eyeless-GFP reporter and also contains the gRNA (gene drive) but not the Cas9 (considered as a safe split gene drive). 'Homing' of the eye-GFP is assayed to detect insertion at the homologous locus when Cas9 is present by HDR.

The authors show that this works well in the female germline with various tested Cas9 lines (vas, nos, Act5C and ubiq-Cas9). In all cases close to 100% transmission to the homologous locus on the homologous chromosome is achieved when an effective guide RNA is used. Hence, eye GFP transmits (“homes”) in a “super-Mendelian” ratio at the chosen target. A male specific transmission works less well (exuL-Cas9).

The reason why it works well appears to be that the chosen target is an essential gene (*Pol-γ35*) in which small changes caused by NHEJ that result in homing “resistant” alleles will be loss of function alleles and hence will not spread in the population.

Unfortunately, the authors did not test, how the drive could spread in a wild type population (no Cas9 expression). I am also missing a test relevant for pest studies that would achieve the spread of a potentially deleterious or beneficial insertion that could kill a population or make it resistant to a disease.

1) This paper is very hard to read. Sentences are excessively long and complicated. References to the Figures appear not always correct.

2) Figure 1. Genotypes in Figure 1A are unreadable in the print version because of the small font. Are the 2 crossing schemes required that only differ in gRNA1 or gRNA2? The surviving progeny should be quantified as in Figure 1B. Figure 1B shows nos-Cas9 and not act-Cas9 results (several typos in subsection “Design and testing of gRNAs targeting an essential gene”).

Figure 1C: the incidence of heterozygous, homozygous and “resistant” cells is schematic and not supported by data, hence questionable if Figure 1C should be shown in results.

3) Figure 2. Genotypes not readable in print. Is it necessary to show schemes of the procedure how transgenic flies were generated and how the *Pol-γ 35* HomeR were made with all chromosomes detailed (Figure 1D)? This could move to the Materials and methods, as it is standard and we learn not much new.

4) More typos: Subsection “Assessment of germline transmission and cleavage rates”: Figure 2B is the wrong reference; Actic 5C should read Actin 5C. Figure 4B GGG codes for Gly (not Gla). sixth paragraph of the Discussion should refer to Figure 6?

5) Figure 5 – as Figure 1 Figure 2, only readable on the computer.

6) It would be interesting to see how the gene drive would spread if Home R and Cas9 would be introduced in a competitive way into wild type populations. This is similar to Figure 4C, but the only the Home R males or females would carry the Cas9. This would be a more realistic test how the gene drive could spread in a wild population that obviously does not express Cas9.

Reviewer #2:

Kandul et al., present an interesting study that could lead to important improvements on the use of homing-based gene drives. However, before publication can be supported there are a number of things that should be addressed to improve the manuscript for better comprehension by readers.

Overall the manuscript presents a load of data. But the presentation of these data could be made in a better digestible way. The authors should go over their maunscript with a reader in mind, that is interested but not necessarily knows all the relevant literature in the very detail.

Abstract: Please remove "inherently confinable" from the Abstract. The drive is indeed designed in a split drive design, however, all the experiments were done in a homozygous Cas9 background. Therefore, there are no experimental data for a split drive provided in this manuscript. The split situation seems to be here more of a practical reason to be allowed to do the experiments in a less stringent laboratory environment. Thus there are no experimental data that would support the confineable nature of this drive. Actually there are not even modelling data to this. Thus, such a statement should not be put in the Abstract. This manuscript is not a demonstration of a confineable drive.

Results: How was *Pol-γ35* identified? It would be interesting to the reader to get to know about the exact reasoning, why this gene was chosen. Or were there several ones chosen before and this turned out to work the best or was the easiest to design. This could be very interesting considerations important to the field.

Results (Figure 1B; Figure 1C) and Materials and methods and Figure 1 (both Figure and legend):

The addressing of the Figure panels and the writing to it don't fit. Has there been a rearrangement of the Figure that was not worked through the text?

When referring to "B" in the text, it is still about Act5C-Cas9 and the nos-Cas9 data are in the text referred to Figure 1C. But Figure 1C is BLM.

In current panel Figure 1B, what does "all" mean below the X-axis? This is not comprehensible. Panel C is not really described in the Figure legend.

Results, Discussion, and Figure 1—figure supplement 4 legend. "converting recessive non-functional resistant alleles into dominant deleterious /lethal mutations" is completely misleading. There is no "conversion" and how should that be done molecularly. There is a continuous removal of such alleles from the population because of lethal transheterozygous conditions caused in the drive. However, there is no active conversion of such alleles into dominant lethal ones. This needs to be clearly rewritten to avoid the misleading idea.

Figure 1—figure supplement 4 also seems to have a slight conceptional problem. What are "cells" (rectangles) with a red frame and a green core? Green means at least one wt allele (this must include the recoded rescue allele.). Red means biallelic knock-out: thus a red cell cannot have a wt allele. Thus what is a red-framed green core cell?

To explain the removal of R2 alleles, a depiction of yellow framed red core cells in the germ line would be helpful, since this would explain how R2 alleles are selected against and might be continuously removed from the population.

Results: Before going into the modelling, the reader should be clearly informed about all the different approaches that are now to be compared. This is currently not done well, if at all. Thus moving current Figure 6 before current Figure 5 might clearly help. Also a better explanation of the panels in Figure 6 is necessary as well as a correction of Fig6 Panel E.

A comparison of a great number of the currently approached toxin-antidote (gene destruction – rescue, but not killer-rescue.) systems is greatly appreciated. However, the authors cannot expect the general reader to know about the small detailed differences between the systems that are compared here. Thus the authors need to do some explanation and categorization of the different approaches here and also cite all the respective literature.

– First subdivision: Non-homing (interference-based drives) VERSUS Homing (thus overreplication-based drives). This will also help then to better understand, why the interference-based drives (TARE and ClvR) are more sensitive to fitness parameters than overreplication drives.

– Second subdivision: same-site VERSUS distant site. This is important to understand the difference between the here modelled TARE and the CLvR. Actually ClvR is a TARE, but you use TARE here more specifically as the results in the respective paper are demonstrating only a same-site TARE. But this needs to be clearly stated here.

– Third subdivision: viable VERSUS haplosufficient VERSUS haploinsufficient. This also needs to be clearly depicted in labellling panels C to F of Figure 6, which are currently hard to grasp what the essential differences are, before looking at the panels in detail:

C: HGD of viable gene (HGD)

D: HGD of viable gene with rescue (HGD+R)

E: HGD of haploinsufficient gene with rescue (HGD-hi+R). This panel needs major correction.

F: HGD of haplosufficient (essential) gene with rescue (HomeR)

– Fourth subdivision: split VERSUS non-split. Here for the split HGD situation, the respective papers of which the current authors are co-authors should be cited: Kandul et al., 2020 and Li et al. 2020. In addition, it is also important to state clearly that "split or two locus" is completely independent of the "distant site" concept.

The reader needs to understand the differences of the systems that are compared here, without having the reader to go to the respective publications themselves and then try to find out what the differences really are. This is not so obvious and the current authors have a clear chance here to do that and help the reader in the mists of all this similar but still distinct approaches.

Figure 6 Panel E: This depiction is not consistent within itself, not consistent with the legend, and not consistent with the cited literature.

– Why should the rescuing drive construct over the wt allele be lethal as indicated in the right two boxes?

– The cited paper Champer et al., 2020b clearly states that there is maternal carry over, which actually makes it so hard to use and is probably only working via male propagation. In the Figure legend it is said that "maternal carryover and somatic expression.… are empirically unavoidable", which is contrast to the depiction. The legend then also states that this is "unachievable". This should be better replaced by "hard to achieve", since the approach is published and seems to drive, even though probably just via the males. Thus the depiction of panel E needs to be thoroughly revised.

Discussion: The haplolethal HGD works (admittingly poorly) despite the maternal carryover (Champer et al., 2020b). Therefore, your statement needs to be refined or deleted: "requires germline-specific promoter that lacks maternal carryover" is not consistent with the published paper. The drive could go via the males because then you do not have maternal carry over. And homing based drives can go via males and do not necessarily have to be promoted through females, see also KaramiNejadRanjbar et al., 2018.

Discussion. This sentence is based on an old but clearly overruled idea. NHEJ repair is not restricted to a time before the fusion of the paternal and maternal genetic material. It has been clearly demonstrated that R1 and R2 alleles are generated in the early embryo also after the zygote state (Champer et al., 2017, KaramiNejadRanjbar et al., 2018). Actually, all of the authors' Figure 1C and Figure 1—figure supplement 4 are about NHEJ mutation in the early embryo causing "BLM". Thus this sentence is inconsistent with current believes and also with the authors' own writing.

Figure 4: Panel C graph: Why is in the controls the transgene consistently and significantly higher inherited to the next generation (0). It is about 75% progeny sired by the transgenic fathers compared to the wild type fathers? Was there an age advantage of the transgenic ones or whatever other fitness factor? This is surprising and no explanation is given at all.

In contrast, in the Cas9 background, in generation 0 less than 50% carry the drive allele, which is probably due to induced lethality. But this should also be commented on.

In the legend it is stated that 7 of 9 flies carried an R1 allele heterozygous to an R2 allele. What about the other two?

Reviewer #3:

The authors are to be commended for the effort put into careful experimental design and clear presentation of methods and results.

My main concern with the manuscript is that the claim about their specific polymerase gene being "ultraconserved" is not backed up with their own data or by citations from the literature. If the gene sequence was ultra-conserved, I wouldn't have expected the authors to be able to do so much recoding of the gene without fitness consequences. Furthermore, it is clear that homozyogous-viable NHEJ mutations did develop in the experiment. Without explanation, this seems to be a fatal flaw in the design.

This manuscript describes a modification of the general homing gene drive concept by use of a split drive system that increases the frequency of a recoded polymerase gene that replaces a cleavage susceptible, naturally occurring, haplosufficient, conserved polymerase gene. This approach is taken in order to limit the evolution of cleavage resistance in the naturally occurring gene.

As mentioned in the summary, I am not convinced that the research presented achieves the intended goals. I did a quick look for literature on the "ultraconserved" polymerase pol-y35 gene a could find none. I am not sure if the conservation is at the DNA sequence level or at the amino acid level. If at the amino acid level, then it makes sense that resistance alleles can form at the DNA level that don't impact the protein at all. Figure 2A shows the 22 and 27 recoded nucleotides for the two guide RNA sites. The authors say that these changes to the sequences didn't seem to impede fitness. Did the authors try many other recodings and finally decide on these because all others caused loss of fitness, or is it just that this gene is robust to substitutions even though the protein is conserved.

Figure 4C shows that the frequency of flies with at least one copy of the pol-y35home R1 increased from about 25% to about 50% between the parental and F0 generation when there was no Cas9 present. As long as the transgenic males were competitive with the wild flies this makes sense because the released flies were homozygous for that allele and the offspring should all have inherited one copy of the gene. What doesn't make sense is that when the work was done with all flies harboring the Cas9, the pol-y35home R1 increased less than in the former case, from the parental to generation F0, the frequency of flies with the pol-y35home R1. In some replicates the frequency of such flies didn't increase at all. It should be noted that the parents were always homozygous. This certainly indicates a fitness cost to the flies with a combination of Cas9 and the homing construct.

In this same Figure, results from the model are plotted. It seems like the model assumes no fitness cost because it shows an exact increase from 25% to 50% flies carrying at least on copy of the pol-y35home R1 theoretical construct. In later generations the experimental results outperform the model. Presumably, this model is used to construct Figure 6. This mismatch needs to be addressed in the manuscript.

The fact that in all three replicates of the experiment without Cas9, the F0 is above 50% indicates that something else may be going on that is unrelated to gene drive. It could be due to heterosis between the two slightly different strains of flies. When wildtype males mate with wildtype females, the offspring are more inbred than when a transgenic male mates with a wildtype female. -just a hypothesis.

Reviewer #4:

Gene drives can be used for sustainable control of disease vectors, and there is a need for a different gene drive strategies that can be tailored to the particular species, timescale, and desired spatial spread. Kandul and colleagues present a welcome new addition to the growing number of strategies for gene drive, called HomeR, that combines elements of killer-rescue and homing-based drive to exert spatiotemporal control over its spread, whilst counteracting the rise of resistant mutations. Whilst it is extremely promising, some major claims of this manuscript are inaccurate or unsupported by the evidence. The authors could easily address the most important concerns by expanding their sequencing analysis to better detect and quantify resistant mutations, paying careful attention not to overstress the potential of this drive to mitigate resistance, and by comparing the relative strengths of different drive strategies instead of focussing only on features that are most flattering to the HomeR strategy.

1) The drive release strategy of Figure 4A and 4C are primed to underestimate and potentially mask resistance. In Figure 4A, where the authors search for signs of resistance, the population was seeded with males that were all homozygous for the drive, meaning that 100% of their G0 progeny will inherit it. As the rate of homing is close to 99%, only a small fraction of their G1 could have inherited a non-drive (potentially resistant allele) allele. In a realistic release scenario, resistant alleles will have ample opportunity to be generated and subsequently selected. Though still far from adequate, resistance testing would have been better performed on samples collected from the lower frequency releases in panel C. This experiment should not be used to draw strong conclusions about resistance to pHomeR, but should be used to make broader observations regarding the spread and stability of the construct.

2) The strategy for sampling resistance will obscure almost all resistance in the population, and would fail to detect even a strong selection for it. Flies were only selected for resistance genotyping if they lacked GFP, meaning they carry two non-HomeR alleles (i.e. homozygous for the R1 allele or transheterozygous with another R1/R2/WT). One would expect most resistant alleles to be heterozygous in a population that was seeded with almost complete drive homozygosity. The authors could, and should, have done more to identify and quantify these. Amplicon sequencing was used to sample the full diversity of alleles in a larger pool of individuals (including GFP+ flies) collected at G10, why was this approach not used throughout? By adopting the approach earlier they would have been able to track the changing frequencies of R1 and R2 alleles over time.

3) The impression given in the Figure and main text is that R1 alleles were rare (or entirely absent), when they were not. In spite of the incredible advantage given to the drive, and a bias in sampling method that would mask the presence of resistant alleles, resistance was observed in every generation tested (G2, G3 and G10). The authors claim that because GFP- individuals were not observed in later generations, the resistant alleles had not come under positive selection. This logic is flawed, and indeed their own amplicon sequencing analysis performed on G10 flies revealed several resistant alleles, including an R1 present in 80% of non-drive alleles. The two most frequent mutant alleles detected were in frame, and I do not agree that these are likely to be deleterious recessive (as the authors speculated). These could be functionally resistant mutations. I believe there were many more R1 alleles in heterozygosity with the HomeR allele, these alleles could have been spreading, but were excluded from the genotyping analysis. Could these putative R1 individuals not have been specifically tested to see if they do, or do not confer resistance?

4) The modelling takes a very limited approach to comparing different drive strategies, and by comparing proof-of-principle designs, important differences are obscured. For example, simple modifications that would mitigate resistance are likely to be included in many designs – such as multiplexing gRNAs. The nuances of each design are lost in a discussion focused on the rate of spread, which is largely irrelevant now because all of drives are predicted to spread well.

5) The authors did not discuss the relevance of having performed releases in a population that was already homozygous for Cas9. Do the release experiments and model really suggest the drive could spread if released into an otherwise WT population? I'm not sure the data presented in this manuscript can support that claim.

[Editors’ note: further revisions were suggested prior to acceptance, as described below.]

Thank you for submitting your revised article "A home and rescue gene drive efficiently spreads and persists in populations" for consideration by *eLife*. Your article has been reviewed by Patricia Wittkopp as the Senior Editor, a Reviewing Editor, and two reviewers. The reviewers have opted to remain anonymous.

The reviewers have discussed the reviews with one another and the Reviewing Editor has drafted this decision to help you prepare a revised submission.

We would like to draw your attention to changes in our policy on revisions we have made in response to COVID-19 (https://elifesciences.org/articles/57162). Specifically, when editors judge that a submitted work as a whole belongs in *eLife* but that some conclusions require a modest amount of additional new data, as they do with your paper, we are asking that the manuscript be revised to either limit claims to those supported by data in hand, or to explicitly state that the relevant conclusions require additional supporting data.

This paper describes a new gene drive system that appear to have some advantage over existing systems. It shows that Cas 9 mediated homology directed repair can be used to insert at synthetic rescue gene into an essential gene. Although there are clearly some limitations to the effectiveness of the drive, to a large extent due to the fitness cost of Cas-9, this new strategy will be a useful path to follow in order to thwart evolution of resistance to the drive.

Essential revisions:

The reviewers appreciated the huge efforts you made to address their initial concerns and in particular the release into Cas-9 negative populations, even though the fitness cost of Cas-9 is a real issue that limit applicability of the approach. We therefore ask you to strongly decrease your claims to better reflect the results described, and in particular to change the title of the paper. You might want to mention that mosquitos could be better suited for this type of approach than *Drosophila*. I therefore expect to see an amended manuscript in the very near future where, we sincerely hope, you will have represented the results without un-necessary hype.

Reviewer #1:

I commend the authors for conducting additional experiments that enable assessment of the drive dynamics of their strategy under conditions when the split drive is introduced into a lab population without Cas9 and testing the Cas9 independently. The finding that the Cas9 has a fitness cost explains some of the previous results. This must have been a lot of work, but I think it was worth the effort.

I appreciate that the authors have removed the term "ultra-conserved", but I am still not comfortable with their use of the term "conserved" in relationship to the focus of the manuscript on gene drive. It's not just the term, but the expectation that this will be a stable drive system. Even with the small sample size in the current laboratory experiments (compared to what would be expected for the size of the target population in a field release) mutations arose that seemed to have no fitness consequence in males even as single copies with an LOF copy. Isn't it therefore reasonable to expect mutations to arise due to NHEJ that wouldn't have fitness effects on males and females? Beyond that, wouldn't a natural population be expected to already harbor some genotypes that would be immediately resistant to this drive? The authors should clearly address why they don't expect this problem with using their design outside of the lab.

In Figure 1—figure supplement 4, the authors show amino acid sequences that appear to be consensus sequences. What is important for this paper is understanding how much variation exists in the DNA sequence for the 3' end part of the domain of the gene for *D. melanogaster* and other potential targets of gene drive. At least for *D. melanogaster* there are many sequences available. Such data may also be available for some other pest insects. Before this paper is accepted, I think it behooves the authors to provide information on this issue that could predict whether this drive would really thwart resistance evolution.

I commend the authors for having done quite a bit of work to simplify the presentation, although, as they say, there is a limit to how much simplification can be done.

Reviewer #2:

1) The main point from my last review was considering the significance of this study. I suggested to test if this gene drive can spread in wild-type populations not expressing Cas9. The authors have now included data that test this and find the gene drive does not spread, possibly because expressing Cas9 comes at cost of fitness.

Considering this negative result, I do see limited impact of the presented data as the method does not work in wild populations. This new result contrasts what the authors state at the end of their abstract, that HomeR would work for wild populations. Hence, I feel this paper should be more suitable for a specialized journal. However, I am not a population geneticist. I leave this issue of impact to the other reviewers/editor. I am also not able to judge the usefulness and accuracy of the new simulations presented in Figure 3 and 4, comparing to other methods without doing any experiments.

2) I appreciate that the authors tried to make this paper more readable. However, I feel there is still a long way to go. Several sentences are still excessively long. 2nd sentence in the introduction extends across 9 lines. Fourth last sentence of intro: 11 lines. Many non-standard abbreviations are used throughout the paper (HG, LBM, EJ, GD, MMEJ, HACK .).

Figure 1—figure supplement 1C. The genotypes on the crosses shown are still much very small and hence unreadable without zooming in. Why do the authors need to show 2 identical crossing schemes with the only difference that gRNA#1 or #2 was used? This information could simply be listed in a table or as done in FigS1D. The authors describe in an extremely complicated way in the text the simple fact that expression of gRNA#1PolG2 in the presence of Act-Cas9 is killing flies more effectively than gRNA#2 PolG2.

Fig S2B. Why do we need a figure that shows how to make transgenes in 2 different ways for both HomeR drives? This should be in the methods. There is no discovery shown. In the end, only one line for each HomeR construct in the PolG2 gene is used for the population experiments. Which method was applied to generate this one is not clear. Again, this distracts from the message and makes the paper hard to read.

---

## [Author Response]

[Editors’ note: the authors resubmitted a revised version of the paper for consideration. What follows is the authors’ response to the first round of review.]

The following points must be addressed as they are the two main points the manuscript is trying to make but they are not well supported:– The major claim of the paper, the release of the drive into a wild type population with no Cas9 must be investigated as there is no practical demonstration of a split drive.

We appreciate this suggestion and have now done additional multigenerational population cage experiments (15 in total.), including the HomeR drives into the WT genetic background at two different release rates, and demonstrated that the spread of HomeR is limited by Cas9 – a desired self-limiting safety feature inherent to the split-gene drive design of HomeR.

– There is also no final demonstration of the repression of R1 alleles: The strategy for sampling resistance appears to not be appropriate and must be changed as suggested by reviewer #4.

We have carefully described our results as to the representation of resistant alleles in our multigenerational population cage experiments in which Homer:Cas9 males were released atat 25% into Cas9 background (5 replicates, followed for 10 generations – reaches fixation). We have provided 12 new multigenerational population cage experiments. Moreover, we assessed fitness of R1 alleles sampled from drive populations and found that these non-silent R1 incurred fitness costs on female carriers. Taken together, these 15 drive experiments illustrate that HomeR is stable, can spread and persist in a Cas9 dependent manner. See more specific comments below.

– The paper claims to have used an ultraconserved gene. As so much importance is placed on the drive design overcoming resistance, this must be demonstrated. A gene that is considered as ultraconserved would require that there is little or no nucleotide variation (e.g. dsx in the Kyrou et al.,). Conservation at the protein level does little to protect against synonymous mutants that would constitute resistance. A better justification for this choice and a demonstration that it was indeed the right choice must be presented.

We have removed the term “ultraconserved” and added more detail as to how and why we chose PolG2 (note: the gene’s name was changed during the review process). We have also provided amino acid alignments to demonstrate just how well conserved this target site is across diverse species from Humans – frogs – chickens – mice – insects (Figure 1—figure supplement 1B).

– The data are really presented in a very difficult way and the paper must be extensively revised with a much broader audience in mind.

We have edited and simplified the language of the paper to target a broader audience. We have moved many of the figures to the supplement to help streamline the paper.

– The reviewers did a very thorough review of the paper that should help you to improve it. The individual reviews are included below.

We agree and to be honest this is one of the most constructive sets of reviews on a paper that we have ever received. We tremendously appreciate all the reviewers for their hard work.

Reviewer #1:This paper shows that Cas 9 mediated homology directed repair can be used to insert at synthetic rescue gene into an essential gene, here mitochondrial Pol-γ35 was chosen. The insertion is marked by an eyeless-GFP reporter and also contains the gRNA (gene drive) but not the Cas9 (considered as a safe split gene drive). 'Homing' of the eye-GFP is assayed to detect insertion at the homologous locus when Cas9 is present by HDR.The authors show that this works well in the female germline with various tested Cas9 lines (vas, nos, Act5C and ubiq-Cas9). In all cases close to 100% transmission to the homologous locus on the homologous chromosome is achieved when an effective guide RNA is used. Hence, eye GFP transmits (“homes”) in a “super-Mendelian” ratio at the chosen target. A male specific transmission works less well (exuL-Cas9).The reason why it works well appears to be that the chosen target is an essential gene (Pol-γ35) in which small changes caused by NHEJ that result in homing 'resistant' alleles will be loss of function alleles and hence will not spread in the population.Unfortunately, the authors did not test, how the drive could spread in a wild type population (no Cas9 expression). I am also missing a test relevant for pest studies that would achieve the spread of a potentially deleterious or beneficial insertion that could kill a population or make it resistant to a disease.

We thank this reviewer for this comment. To strengthen this paper, we have now included a total of three separate multigenerational population cage experiments (15 experiments with replicates in total). These include release of Homer:Cas9 males at 50% in Cas9 background (3 replicates, followed for 10 generations – reaches fixation), and release of Homer:Cas9 males at 25% in Cas9 background (5 replicates, followed for 10 generations – reaches fixation). We also perform a negative control experiment by releasing Homer males at 25% in WT background (3 replicates, followed for 10 generations, does not reach fixation as expected since drive is Cas9 dependent and inherently confineable). In addition to these experiments, we also have included additional experimental data illustrating the behavior of the HomeR system in a wildtype population released at two introduction frequencies. These include the release of HomeR:Cas9 males at50% into a WT background (2 replicates, followed for 6 generations) and the release of HomeR:Cas9 males at75% into a WT background (2 replicates, followed for 6 generations). As expected from all these experiments, the HomeR drive persists/spreads in a Cas9 dependent manner, making the drive inherently confineable. Moreover, we have provided further modelling to illustrate the behavior with multi-releases of HomeR drive with varied fitness costs to both the drive and to the Cas9 alleles.

1) This paper is very hard to read. Sentences are excessively long and complicated. References to the Figures appear not always correct.

We thank this reviewer for this comment. We have extensively revised this manuscript and have moved many of the non-essential results (e.g. second gRNA construct details) to the supplement to make the manuscript easier to digest.

2) Figure 1. Genotypes in Figure 1A are unreadable in the print version because of the small font.

We appreciate this comment – we have increased the font/readability of this panel and also moved this Figure to Figure 1—figure supplement 1C.

Are the 2 crossing schemes required that only differ in gRNA1 or gRNA2?

Yes, there are two crossing schemes differing by gRNA (1 or 2). We wanted to illustrate that gRNA#1/Cas9 is lethal to all trans-hets (male/female) while gRNA#2 is lethal only to trans-het males. We hope that by increasing the font size this distinction will now be clearer.

The surviving progeny should be quantified as in Figure 1B. Figure 1B shows nos-Cas9 and not act-Cas9 results (several typos in subsection “Design and testing of gRNAs targeting an essential gene”).

Corrected. The data for both Nos-Cas9 and Act5C-Cas9 can be found in Supplementary fil2 and Supplementary file 3.

Figure 1C: the incidence of heterozygous, homozygous and “resistant” cells is schematic and not supported by data, hence questionable if Figure 1C should be shown in results.

Corrected. We appreciate this suggestion and have moved this panel to Figure 1—figure supplement 4. LBM is a mechanism explaining the data. The expression of Cas9/gRNA during development induces independent mutations in somatic and germ cells resulting in lethality at the organism level.

3) Figure 2. Genotypes not readable in print. Is it necessary to show schemes of the procedure how transgenic flies were generated and how the Pol-γ35 HomeR were made with all chromosomes detailed (Figure 1D)? This could move to the Materials and methods as it is standard and we learn not much new.

Corrected. We appreciate this suggestion and have moved this from the main figures and to Figure 1—figure supplement 2.

4) More typos: Subsection “Assessment of germline transmission and cleavage rates”:Figure 2B is the wrong reference.

Corrected.

Actic 5C should read Actin 5C.

Corrected.

Figure 4B GGG codes for Gly (not Gla).

Corrected, now Figure 2—figure supplement 1.

Sixth paragraph of the Discussion should refer to Figure 6?

Corrected.

5) Figure 5 – as Figure 1 Figure 2, only readable on the computer.

We have updated and modified the figures to make them more readable.

6) It would be interesting to see how the gene drive would spread if Home R and Cas9 would be introduced in a competitive way into wild type populations. This is similar to Figure 4C, but the only the Home R males or females would carry the Cas9. This would be a more realistic test how the gene drive could spread in a wild population that obviously does not express Cas9.

We agree, and we have now included those experiments. In total we now have 15 multi-generational drive experiments. We have also included additional mathematical modelling to support this new data and make predictions related to fitness costs (to wither allele: Cas9 of the Homer) in addition to multi-release scenarios.

Reviewer #2:Kandul et al., present an interesting study that could lead to important improvements on the use of homing-based gene drives. However, before publication can be supported there are a number of things that should be addressed to improve the manuscript for better comprehension by readers.Overall the manuscript presents a load of data. But the presentation of these data could be made in a better digestible way. The authors should go over their manuscript with a reader in mind, that is interested but not necessarily knows all the relevant literature in the very detail.

We thank the reviewer for this comment and have significantly revised the manuscript and moved much of the nonessential material to the methods /supplement to make the paper more digestible.

Abstract: Please remove "inherently confinable" from the abstract. The drive is indeed designed in a split drive design, however, all the experiments were done in a homozygous Cas9 background. Therefore, there are no experimental data for a split drive provided in this manuscript. The split situation seems to be here more of a practical reason to be allowed to do the experiments in a less stringent laboratory environment. Thus there are no experimental data that would support the confineable nature of this drive. Actually there are not even modelling data to this. Thus, such a statement should not be put in the Abstract. This manuscript is not a demonstration of a confineable drive.

We thank the reviewer for this comment. However, we have provided new population cage data (See response to reviewer 1 comments above) demonstrating that Homer spreads in a Cas9 dependent manner. These included multiple experimental releases of HomeR into WT populations demonstrating both drive capacity, persistence and confineblity. Therefore, given this ample new data, we prefer to leave in the terms “inherently confineable,” as we have clearly demonstrated this potential.

Results: How was Pol-γ35 identified? It would be interesting to the reader to get to know about the exact reasoning, why this gene was chosen. Or were there several ones chosen before and this turned out to work the best or was the easiest to design. This could be very interesting considerations important to the field.

We have added in more detail as to why we chose this target gene.

“Selection of *PolG2* as a HomeR drive target.

To develop a HomeR-based drive, we first identified an essential haplosufficient gene to target. We chose *DNA Polymerase γ subunit 2* (*PolG2, DNA polymerase Ɣ 35-kDa,* CG33650), required for the replication and repair of mitochondrial DNA (mtDNA) (Carrodeguas, 2000; Carrodeguas et al., 2001) whose LOF results in lethality (Iyengar et al., 2002). *PolG2* encodes the small subunit of the mitochondrial DNA polymerase γ, acting together with the large subunit 1 (*PolG1, DNA polymerase Ɣ 125-kDa,* CG8987) for the replication and repair of the mitochondrial genome (Carrodeguas, 2000; Carrodeguas et al., 2001). *PolG2* is a short and conserved gene, and its C-terminal domain (cd02426, (Lu et al., 2020)) is roughly 130 amino acids (AA) and shares 55% AA identity with the Human *PolG2 (Lecrenier et al., 1997)* (Figure 1—figure supplement 1A,B). Importantly, *Drosophila PolG2* loss-of-function (LOF) mutations are known to cause lethality at the early pupal stage (Iyengar et al., 2002). The C-terminal location of the functional domain in *PolG2* facilitates its re-coding, making *PolG2* an optimal target for a HomeR gene drive (Figure 1A).*”*

Moreover, we want to point out here that since receiving this paper back from peer review – Flybase has updated the gene name for this gene to PolG2 – and we have therefore modified this gene name throughout the entire manuscript.

Results (Figure 1B; Figure 1C) and Materials and methods and Figure 1 (both Figure and legend):The addressing of the Figure panels and the writing to it don't fit. Has there been a rearrangement of the Figure that was not worked through the text?When referring to "B" in the text, it is still about Act5C-Cas9 and the nos-Cas9 data are in the text referred to Figure 1C. But Figure 1C is BLM.

Corrected. We have updated the figures to correct this issue.

In current panel Figure 1B, what does "all" mean below the X-axis? This is not comprehensible.

Corrected. We have edited the data presentation in this figure panel to make it easier to comprehend. We have also indicated where the raw data for this panel can be found.

Panel C is not really described in the Figure legend.

Corrected. We have removed this panel from this Figure. LBM is now described in Figure 1—figure supplement 4.

Results, Discussion, and Figure 1—figure supplement 4legend. "converting recessive non-functional resistant alleles into dominant deleterious /lethal mutations" is completely misleading. There is no "conversion" and how should that be done molecularly. There is a continuous removal of such alleles from the population because of lethal transheterozygous conditions caused in the drive. However, there is no active conversion of such alleles into dominant lethal ones. This needs to be clearly rewritten to avoid the misleading idea.

Corrected. We appreciate this comment and have revised our wording to better describe the mechanism of BLM.

Figure 1—figure supplement 4 also seems to have a slight conceptional problem. What are "cells" (rectangles) with a red frame and a green core? Green means at least one wt allele (this must include the recoded rescue allele.). Red means biallelic knock-out: thus a red cell cannot have a wt allele. Thus what is a red-framed green core cell?To explain the removal of R2 alleles, a depiction of yellow framed red core cells in the germ line would be helpful, since this would explain how R2 alleles are selected against and might be continuously removed from the population.

Corrected. We appreciate this comment and have revised this figure.

Results: Before going into the modelling, the reader should be clearly informed about all the different approaches that are now to be compared. This is currently not done well, if at all. Thus moving current Figure 6 before current Figure 5 might clearly help. Also a better explanation of the panels in Figure 6 is necessary as well as a correction of Fig6 Panel E.A comparison of a great number of the currently approached toxin-antidote (gene destruction – rescue, but not killer-rescue.) systems is greatly appreciated. However, the authors cannot expect the general reader to know about the small detailed differences between the systems that are compared here. Thus the authors need to do some explanation and categorization of the different approaches here and also cite all the respective literature.

We appreciate this comment and agree that a detailed description of each system would be great. However, we prefer to focus on the results from our study and have therefore directed the reader to Figure 3—figure supplement 1 for mechanistic comparisons of each of these systems. What would be useful at this point would be a detailed review article that covers and compares each of these approaches – perhaps we could take that on later.

– First subdivision: Non-homing (interference-based drives) VERSUS Homing (thus overreplication-based drives). This will also help then to better understand, why the interference-based drives (TARE and ClvR) are more sensitive to fitness parameters than overreplication drives.

This distinction is present in the Figure. See top right. Non-homing (empty Circle) ; Homing (green Circle)

– Second subdivision: same-site VERSUS distant site. This is important to understand the difference between the here modelled TARE and the CLvR. Actually ClvR is a TARE, but you use TARE here more specifically as the results in the respective paper are demonstrating only a same-site TARE. But this needs to be clearly stated here.

We appreciate the suggestion. We have added in same-site (teal diamond) and distant-site (red ellipse) into the schematic. This should help demonstrate that we are depicting a same-site TARE as the reviewer points out.

– Third subdivision: viable VERSUS haplosufficient VERSUS haploinsufficient. This also needs to be clearly depicted in labellling panels C to F of Figure 3—figure supplement 1, which are currently hard to grasp what the essential differences are, before looking at the panels in detail:

This distinction is present in the Figure. See top right. Non-essential gene (C,D; Yellow Hexagon); Haplosufficient (F; Orange Square) ; Haploinsufficeint (E; blue triangle)

C: HGD of viable gene (HGD)D: HGD of viable gene with rescue (HGD+R)E: HGD of haploinsufficient gene with rescue (HGD-hi+R). This panel needs major correction.F: HGD of haplosufficiant (essential) gene with rescue (HomeR)

Corrected. We appreciate these suggestions – and have updated our Figure for more clarity. All of this information can now be found in the figure itself or the Figure legend.

– Fourth subdivision: split VERSUS non-split. Here for the split HGD situation, the respective papers of which the current authors are co-authors should be cited: Kandul et al., 2020 and Li et al. 2020. In addition, it is also important to state clearly that "split or two locus" is completely independent of the "distant site" concept.

Corrected. All the drives in this Figure are split designs and we have added that descriptor into the title “Mechanistic comparison of contemporary split-drives for population modification.” If we wanted to compare non-split – we would need to generate a new figure and given that HomeR is a split system we don’t think that is necessary here. Those papers (Kandul et al., 2020 and Li et al., 2020) have been cited.

The reader needs to understand the differences of the systems that are compared here, without having the reader to go to the respective publications themselves and then try to find out what the differences really are. This is not so obvious and the current authors have a clear chance here to do that and help the reader in the mists of all this similar but still distinct approaches.

We have tried our best to articulate these systems and make clear what the differences are. This underscores that what is really needed is a detailed review on this topic that covers these various systems. Please reach out to us if you might be interested in writing that piece with us in the future.

Figure 6 Panel E: This depiction is not consistent within itself, not consistent with the legend, and not consistent with the cited literature.– Why should the rescuing drive construct over the wt allele be lethal as indicated in the right two boxes?

Thank you for pointing this. We have modified the figure legend to match the depiction in panel E. The reviewer asks, “Why should the rescuing drive construct over the wt allele be lethal as indicated in the right two boxes?” This results from maternal deposition/somatic expression of Cas9/gRNA acting on the “WT” allele resulting in disruption of that allele. Given that this is targeting a Haploleathal gene – having only one functional copy is not sufficient – hence lethality. From our experiences using the Nanos-Cas9 line, which is the same line used by Champer et al., 2020, we find significant maternal deposition and somatic activity. That said, the reviewer is correct, Champer et al., 2020 reportedly had “minimal somatic expression” from Nanos-Cas9 and therefore we have updated this figure to reflect these findings.

– The cited paper Champer et al., 2020b clearly states that there is maternal carry over, which actually makes it so hard to use and is probably only working via male propagation. In the Figure legend it is said that "maternal carryover and somatic expression.… are empirically unavoidable", which is contrast to the depiction. The legend then also states that this is "unachievable". This should be better replaced by "hard to achieve", since the approach is published and seems to drive, even though probably just via the males. Thus the depiction of panel E needs to be thoroughly revised.

Corrected.

Discussion: The haplolethal HGD works (admittingly poorly) despite the maternal carryover (Champer et al., 2020b). Therefore, your statement needs to be refined or deleted: "requires germline-specific promoter that lacks maternal carryover" is not consistent with the published paper. The drive could go via the males because then you do not have maternal carry over. And homing based drives can go via males and do not necessarily have to be promoted through females, see also KaramiNejadRanjbar et al., 2018.

We thank the reviewer for this comment and have modified this sentence.

Discussion. This sentence is based on an old but clearly overruled idea. NHEJ repair is not restricted to a time before the fusion of the paternal and maternal genetic material. It has been clearly demonstrated that R1 and R2 alleles are generated in the early embryo also after the zygote state (Champer et al., 2017, KaramiNejadRanjbar et al., 2018). Actually, all of the authors' Figure 1C and Figure 1—figure supplement 4 are about NHEJ mutation in the early embryo causing "BLM". Thus this sentence is inconsistent with current believes and also with the authors' own writing.

We thank the reviewer for this comment and have modified this sentence.

Figure 4: Panel C graph: Why is in the controls the transgene consistently and significantly higher inherited to the next generation (0). It is about 75% progeny sired by the transgenic fathers compared to the wild type fathers? Was there an age advantage of the transgenic ones or whatever other fitness factor? This is surprising and no explanation is given at all.In contrast, in the Cas9 background, in generation 0 less than 50% carry the drive allele, which is probably due to induced lethality. But this should also be commented on.

We thank the reviewer for pointing this out. We have mentioned the higher mating competitiveness of HomeR without Cas9 relative to WT males in the text (Figure 2B). This data suggests that HomeR provides a good rescue and does not incur fitness costs without Cas9. We now clearly show that nos-Cas9 does incur fitness costs to its carriers alone (Figure 2C), so then HomeR+Cas9 males released together with Cas9-alone male, the former are less competitive (Figure 2B).

In the legend it is stated that 7 of 9 flies carried an R1 allele heterozygous to an R2 allele. What about the other two?

We thank the reviewer for this comment and have clarified these results in the manuscript. The current legend of Figure 3—figure supplement 1: “Seven out of nine flies were heterozygous, harboring one of the identified *PolG2^R1^* alleles together with an out-offrame *indel* (LOF) allele. The remaining two GFP- flies were likely homozygous for the R1#1 allele, because ten randomly sequenced clones harbored the same allele.” We described in the method section that PCR amplicons from a single fly were subcloned into the pCR2.1-TOPO plasmid, and at least 7 clones were sequenced in both directions by Sanger sequencing. In general, we were able to identify both allele from sequencing 7 clones. If all 7 clones were identical provide, we would sequence 3 additional clones.

Reviewer #3:The authors are to be commended for the effort put into careful experimental design and clear presentation of methods and results.My main concern with the manuscript is that the claim about their specific polymerase gene being "ultraconserved" is not backed up with their own data or by citations from the literature. If the gene sequence was ultra-conserved, I wouldn't have expected the authors to be able to do so much recoding of the gene without fitness consequences. Furthermore, it is clear that homozyogous-viable NHEJ mutations did develop in the experiment. Without explanation, this seems to be a fatal flaw in the design.

Corrected. We thank the reviewer for this comment and have removed the term “ultraconserved” from the manuscript. We have also expanded the discussion on resistant alleles that were viable, and assess their fitness. It should be noted that every drive system is susceptible to resistant alleles and these cannot be 100% avoided by any design. We have not identified any silent R1 allele: every sampled in-frame resistant alleles changes the amino acid sequence of PolG2.

This manuscript describes a modification of the general homing gene drive concept by use of a split drive system that increases the frequency of a recoded polymerase gene that replaces a cleavage susceptible, naturally occurring, haplosufficient, conserved polymerase gene. This approach is taken in order to limit the evolution of cleavage resistance in the naturally occurring gene.As mentioned in the summary, I am not convinced that the research presented achieves the intended goals. I did a quick look for literature on the "ultraconserved" polymerase pol-y35 gene a could find none. I am not sure if the conservation is at the DNA sequence level or at the amino acid level. If at the amino acid level, then it makes sense that resistance alleles can form at the DNA level that don't impact the protein at all. Figure 2A shows the 22 and 27 recoded nucleotides for the two guide RNA sites. The authors say that these changes to the sequences didn't seem to impede fitness. Did the authors try many other recodings and finally decide on these because all others caused loss of fitness, or is it just that this gene is robust to substitutions even though the protein is conserved.

We thank the reviewer for this comment, the term “ultraconserved” was removed from the manuscript. In terms of the fitness question, we did not observe any significant decreases in fitness. Moreover, when Homer was released at 25% allele frequency – it remained relatively stable in 3 multigenerational population cage experiments observed for 10 generations (Figure 2B). Additionally, we targeted the 3’ end of the gene to minimize the degree of recoding required to further minimize potential fitness impacts. To help illustrate the degree of conservation we are referring to – we have provided the sequence alignments in Figure 1—figure supplement 1B.

Figure 4C shows that the frequency of flies with at least one copy of the pol-y35home R1 increased from about 25% to about 50% between the parental and F0 generation when there was no Cas9 present. As long as the transgenic males were competitive with the wild flies this makes sense because the released flies were homozygous for that allele and the offspring should all have inherited one copy of the gene. What doesn't make sense is that when the work was done with all flies harboring the Cas9, the pol-y35home R1 increased less than in the former case, from the parental to generation F0, the frequency of flies with the pol-y35home R1. In some replicates the frequency of such flies didn't increase at all. It should be noted that the parents were always homozygous. This certainly indicates a fitness cost to the flies with a combination of Cas9 and the homing construct.

We appreciate this comment and agree that there seems to be a major fitness cost when Cas9 is present. In this revision we have provided additional multigenerational population cage experiment data to support this claim. For example, Figure 2C shows HomeR:Nos-Cas9 male releases at two thresholds (25% and 37.5% allele frequencies) into a WT background. In the first couple generations we can see a bit of stochasticity – however at generation 2-onward we see the Nos-Cas9 allele rapidly decline in frequency – heading toward elimination. This illustrates the inherent fitness cost the Cas9 allele carries which can explain the discrepancy this reviewer highlights.

In this same Figure, results from the model are plotted. It seems like the model assumes no fitness cost because it shows an exact increase from 25% to 50% flies carrying at least on copy of the pol-y35home R1 theoretical construct. In later generations the experimental results outperform the model. Presumably, this model is used to construct figure 6. This mismatch needs to be addressed in the manuscript.

Our original fits indicated that males experienced very few issues due to expression of Cas9, leading to the near perfect inheritance seen by the reviewer. However, we believe that was an artifact of a Cas9 population, and have since updated our work to reflect the new cage experiments, current Figure 3C, HomeR:Nos-Cas9 released into a WT background. These experiments allowed us to uncover a significant mating reduction in males, due to expression of Cas9, which is also reflected in HomeR-Exp. simulations in figure 5A and 5C. This fitness cost aligns our model with the current experimental work. The model does indeed take into account finess costs and these are more impactful later in the drive spread – but it does not take into account mating competition / inbreeding.

The fact that in all three replicates of the experiment without Cas9, the F0 is above 50% indicates that something else may be going on that is unrelated to gene drive. It could be due to heterosis between the two slightly different strains of flies. When wildtype males mate with wildtype females, the offspring are more inbred than when a transgenic male mates with a wildtype female. -just a hypothesis.

Heterosis is an interesting hypothesis. We think this is more likely due to stochasticity resulting from small caged populations in addition to the fitness costs associated with Cas9 as articulated above. Regardless, we have provided ample additional multigenerational population data (15 in total) further supporting the conclusion that HomeR can spread in a Cas9 dependent manner.

Reviewer #4:Gene drives can be used for sustainable control of disease vectors, and there is a need for a different gene drive strategies that can be tailored to the particular species, timescale, and desired spatial spread. Kandul and colleagues present a welcome new addition to the growing number of strategies for gene drive, called HomeR, that combines elements of killer-rescue and homing-based drive to exert spatiotemporal control over its spread, whilst counteracting the rise of resistant mutations. Whilst it is extremely promising, some major claims of this manuscript are inaccurate or unsupported by the evidence. The authors could easily address the most important concerns by expanding their sequencing analysis to better detect and quantify resistant mutations, paying careful attention not to overstress the potential of this drive to mitigate resistance, and by comparing the relative strengths of different drive strategies instead of focussing only on features that are most flattering to the HomeR strategy.1) The drive release strategy of Figure 4A and 4C are primed to underestimate and potentially mask resistance. In Figure 4A, where the authors search for signs of resistance, the population was seeded with males that were all homozygous for the drive, meaning that 100% of their G0 progeny will inherit it. As the rate of homing is close to 99%, only a small fraction of their G1 could have inherited a non-drive (potentially resistant allele) allele. In a realistic release scenario, resistant alleles will have ample opportunity to be generated and subsequently selected. Though still far from adequate, resistance testing would have been better performed on samples collected from the lower frequency releases in panel C. This experiment should not be used to draw strong conclusions about resistance to pHomeR, but should be used to make broader observations regarding the spread and stability of the construct.

We agree with this reviewer and appreciate this comment. The experiments in Figure 4A (new Figure 2A) were designed to mostly explore the stability of HomeR and “fish out” any frequent R1 resistant allele(s) induced in generation 0. In this new revision we have provided ample new multigenerational population cage experiment data – assessing the performance of HomeR when seeded at lower frequencies (e.g. 25% and 37%). These lower release frequencies have indeed provided opportunities for resistant alleles to be generated and selected at the expense of the HomeR allele, this did not happen. In the presence of Cas9, HomeR spread to the fixation (new Figure 2B). The Drives in new Figure 2C also spread or were able to persist at moderate frequency – but the Cas9 allele had too high of a fitness cost and quickly fell out of the populations – limiting the drive spread ability. That said, we did sample additional R1 resistant alleles from the drives in new Figure 2B – indicating again that these alleles can indeed be generated. However, we crudely assessment of fertility of the sampled R1 alleles before genotyping them; and found the sampled R1 alleles imposed fitness cost to the HomeR- viable females.

2) The strategy for sampling resistance will obscure almost all resistance in the population, and would fail to detect even a strong selection for it. Flies were only selected for resistance genotyping if they lacked GFP, meaning they carry two non-HomeR alleles (i.e. homozygous for the R1 allele or transheterozygous with another R1/R2/WT). One would expect most resistant alleles to be heterozygous in a population that was seeded with almost complete drive homozygosity. The authors could, and should, have done more to identify and quantify these. Amplicon sequencing was used to sample the full diversity of alleles in a larger pool of individuals (including GFP+ flies) collected at G10, why was this approach not used throughout? By adopting the approach earlier they would have been able to track the changing frequencies of R1 and R2 alleles over time.

We agree with the reviewer that by sequencing every generation would be an ideal approach to assess the fate of all resistant alleles, and would be extremely beneficial for the gene drive field. However, it was beyond the scope and goal of this project. This would be a large, very interesting, and expensive project endeavour, that we would prefer to save for work in an actual pest species (i.e. mosquitoes). Illumina amplicon sequencing was undertaken primarily to see if the R1 alleles identified in generations 2 and 3 persisted until generation 10. Just as a note, we think that due to the HomeR design R2 alleles are not going to block its spread, they can only slow its genotypic fixation. Whereas R1 alleles can block the spread of any homing based gene drive system, including HomeR. We understand that our approach with 320 heterozygous flies at generation 0 is limited -- R1 alleles are hidden by HomeR alleles. WT alleles in heterozygous females will be converted into HomeR alleles, but many WT alleles in heterozygous males will be passed to the next generation as WT, R1 or R2 alleles. Moreover, we have provided additional multigenerational population cage experiments to support the drive propensity, stability and confineability of HomeR. We have also sampled additional R1 resistant alleles from these cage experiments and crudely assess the fitness of their carriers.

3) The impression given in the Figure and main text is that R1 alleles were rare (or entirely absent), when they were not. In spite of the incredible advantage given to the drive, and a bias in sampling method that would mask the presence of resistant alleles, resistance was observed in every generation tested (G2, G3 and G10). The authors claim that because GFP- individuals were not observed in later generations, the resistant alleles had not come under positive selection. This logic is flawed, and indeed their own amplicon sequencing analysis performed on G10 flies revealed several resistant alleles, including an R1 present in 80% of non-drive alleles. The two most frequent mutant alleles detected were in frame, and I do not agree that these are likely to be deleterious recessive (as the authors speculated). These could be functionally resistant mutations. I believe there were many more R1 alleles in heterozygosity with the HomeR allele, these alleles could have been spreading, but were excluded from the genotyping analysis. Could these putative R1 individuals not have been specifically tested to see if they do, or do not confer resistance?

We removed any indication that these R1 alleles were rare events since they were observed at each generation tested. Moreover, we removed the logic that GFP- individuals were not observed in later generations, the resistant alleles had not come under positive selection. We also added a note into the description of this data indicating the limitation of this analysis: “we cannot rule out the possibility that there were many other potential resistant alleles that were present in the population – that could have been masked by the HomeR allele.”

4) The modelling takes a very limited approach to comparing different drive strategies, and by comparing proof-of-principle designs, important differences are obscured. For example, simple modifications that would mitigate resistance are likely to be included in many designs – such as multiplexing gRNAs. The nuances of each design are lost in a discussion focused on the rate of spread, which is largely irrelevant now because all of drives are predicted to spread well.

While we appreciate the reviewer’s comments, we consider such “proof-of-principle” constructs demonstrative of any mitigating modifications to these designs. We also think that the rate of gene drive spread is an important parameter since the slower spread can be blocked by the immigration of existent R1 alleles from neighboring populations. Moreover, we have provided additional modelling to this manuscript taking into consideration multi-releases, fitness and transmission rates. We hope these models will help the reader understand and compare the HomeR system to other contemporary drive designs (see response to #5 below).

5) The authors did not discuss the relevance of having performed releases in a population that was already homozygous for Cas9. Do the release experiments and model really suggest the drive could spread if released into an otherwise WT population? I'm not sure the data presented in this manuscript can support that claim.

The reviewer is correct, we did not provide experimental data of HomeR releases into a WT population in our first version of this paper. That said, this revision includes new multigenerational population cage experiments (15 experiments in total – this was A LOT of work) demonstrating HomeR’s drive potential, stability and confineability. Additionally, we have provided additional mathematical models, determining the required number of releases of HomeR, into a naive population, to reach a population frequency of 95%. These simulations implement overlapping generations and density-dependent effects in larval stages (reflecting biological conditions), along with drive efficacy and fitness parameters estimated from the new cage trails, to accurately estimate the behavior of HomeR in the field.

[Editors’ note: what follows is the authors’ response to the second round of review.]

Reviewer #1:I commend the authors for conducting additional experiments that enable assessment of the drive dynamics of their strategy under conditions when the split drive is introduced into a lab population without Cas9 and testing the Cas9 independently. The finding that the Cas9 has a fitness cost explains some of the previous results. This must have been a lot of work, but I think it was worth the effort.

Thank you for this feedback. We agree, this was a significant amount of work – but it was essential to include this additional data.

I appreciate that the authors have removed the term "ultra-conserved", but I am still not comfortable with their use of the term "conserved" in relationship to the focus of the manuscript on gene drive. It's not just the term, but the expectation that this will be a stable drive system. Even with the small sample size in the current laboratory experiments (compared to what would be expected for the size of the target population in a field release) mutations arose that seemed to have no fitness consequence in males even as single copies with an LOF copy. Isn't it therefore reasonable to expect mutations to arise due to NHEJ that wouldn't have fitness effects on males and females? Beyond that, wouldn't a natural population be expected to already harbor some genotypes that would be immediately resistant to this drive? The authors should clearly address why they don't expect this problem with using their design outside of the lab.

We appreciate this comment and have removed the word conserved throughout the manuscript. We stressed further in the revised version of the manuscript that some functional resistant alleles can exist in natural populations and together with induced functional resistance alleles they can slow down the spread of HomeR and hinder its spread. We agree with the reviewer that eventually functional resistant alleles can be sampled or induced, and no homing gene drive system is completely stable in the face of natural selection. We offer suggestions on how to improve HomeR drive stability by multiplexing gRNAs.

In Figure 1—figure supplement 4, the authors show amino acid sequences that appear to be consensus sequences. What is important for this paper is understanding how much variation exists in the DNA sequence for the 3' end part of the domain of the gene for *D. melanogaster* and other potential targets of gene drive. At least for *D. melanogaster* there are many sequences available. Such data may also be available for some other pest insects. Before this paper is accepted, I think it behooves the authors to provide information on this issue that could predict whether this drive would really thwart resistance evolution.

Thank you, this is a very great suggestion. We previously sequenced the targeted area in PolG2 in a few *D. melanogaster* lines available in the lab (Oregon, Canton S, etc.), but did not explore the SNP datasets available for *D. melanogaster*. To explore these datasets, we downloaded the *Drosophila melanogaster* Genetic Reference Panel 2 (http://dgrp2.gnets.ncsu.edu) that includes natural variation in genome architecture among 205 *Drosophila melanogaster* Genetic Reference Panel lines (Mackay et al., 2012; Huang et al., 2014) to search for SNPs mapped to the target sequences. Interestingly, we did not find any SNPs mapped to the target sequences. The nearest SNP is located 8 bases downstream from the PAM sequence of gRNA#1 (the blue C in Author response image 1). A description of this analysis was added to the revised manuscript. Author response image 1 shows the alignment of 63 bases including both gRNAs of 13 *Drosophila* species (consensus) sequences.

**Author response image 1. sa2fig1:** 

I commend the authors for having done quite a bit of work to simplify the presentation, although, as they say, there is a limit to how much simplification can be done.

Thank you for this feedback.

Reviewer #2:1) The main point from my last review was considering the significance of this study. I suggested to test if this gene drive can spread in wild-type populations not expressing Cas9. The authors have now included data that test this and find the gene drive does not spread, possibly because expressing Cas9 comes at cost of fitness. Considering this negative result, I do see limited impact of the presented data as the method does not work in wild populations. This new result contrasts what the authors state at the end of their abstract, that HomeR would work for wild populations. Hence, I feel this paper should be more suitable for a specialized journal. However, I am not a population geneticist. I leave this issue of impact to the other reviewers/editor. I am also not able to judge the usefulness and accuracy of the new simulations presented in Figure 3 and 4, comparing to other methods without doing any experiments.

We have described a novel design of the homing gene drive (HomeR) and thoroughly showed that it spreads via homing by scoring the egg-to-adult rate. The assessment of egg-to-adult rate has only recently become a golden standard for assessment of drive mechanism. We have built HomeR as a split-drive (two-locus) to engineer a biocontainment (safety feature) into its design. The HomeR drive is dependent upon Cas9. When Cas9 levels are high in a population the HomeR spreads to fixation (Figure 2A,B). When Cas9 is introduced at low frequency (25%) and continues to reduce in frequency in subsequent generations (nearing 0 by Gen 6), resulting from inherent fitness costs, HomeR spreads in the first few generations (while Cas9 is higher in frequency) then persists (above >50% frequency in all 4 independent populations tested). This should not be misinterpreted as “ authors have now included data that test this and find the gene drive does not spread, possibly because expressing Cas9 comes at cost of fitness.” This result demonstrates that HomeR can spread and its spread is limited by the availability of Cas9. This result was expected, since it is a biocontainment safety switch incorporated into a split-drive design, and is not a “negative result.” Moreover, it should be noted that we only used one release threshold here, and by providing mathematical modeling we can see that HomeR would perform well under a multi-release scenario (Fig 4) against a WT population. Limiting the spread of a split-drive has been a desirable outcome - and a HomeR drive performed well for this purpose in a WT population.

2) I appreciate that the authors tried to make this paper more readable. However, I feel there is still a long way to go. Several sentences are still excessively long. 2nd sentence in the introduction extends across 9 lines. Fourth last sentence of intro: 11 lines. Many non-standard abbreviations are used throughout the paper (HG, LBM, EJ, GD, MMEJ, HACK .).

We appreciate this feedback. These sentences have been corrected. In terms of non-standard abbreviations, unfortunately these are all terms that have been used throughout the literature to describe processes related to DNA repair: End Joining (EJ), Microhomology Mediated End Joining (MMEJ), loss-of-function (LOF) alleles, Homology assisted CRISPR Knock-In (HACK), or used throughout the gene drive literature; Homing Gene Drive (HGD) and Lethal Biallelic Mosaicism (LBM). In fact, none of the terms listed are original to this paper and have been adopted from other published work. Therefore, these are standard abbreviations used in this field.

Figure 1-figure supplement 1C. The genotypes on the crosses shown are still much very small and hence unreadable without zooming in. Why do the authors need to show 2 identical crossing schemes with the only difference that gRNA#1 or #2 was used? This information could simply be listed in a table or as done in FigS1D. The authors describe in an extremely complicated way in the text the simple fact that expression of gRNA#1PolG2 in the presence of Act-Cas9 is killing flies more effectively than gRNA#2 PolG2.

We appreciate this feedback and have increased the font size of Figure 1-figure supplement 1C. We opted to show both crossing schemes as we wanted to be clear which genotypes are perishing. Given that this is a supplemental figure and we are not limited by space - we prefer to keep this figure as is. We have modified the text to make it less complicated as the reviewer suggests.

Fig S2B. Why do we need a figure that shows how to make transgenes in 2 different ways for both HomeR drives? This should be in the methods. There is no discovery shown. In the end, only one line for each HomeR construct in the PolG2 gene is used for the population experiments. Which method was applied to generate this one is not clear. Again, this distracts from the message and makes the paper hard to read.

We appreciate this comment and have added the relevant information to the methods. We thank that the method may be difficult for some readers and hence have made this figure a supplemental figure. We think this figure is important to help readers understand the methods we used to generate the gene drive lines and therefore would prefer to keep this supplemental figure.